# *Desmodium* volatiles in 'push-pull' cropping systems and protection against the fall armyworm, *Spodoptera frugiperda*

Daria M Odermatt[1,2,3]*, Frank Chidawanyika[4,5], Daniel M Mutyambai[4,6], Bernhard Schmid[1], Luiz A Domeignoz Horta[3,7], Collins O Onjura[4], Amanuel Tamiru[4], Meredith C Schuman[1,2]*

[1]University of Zurich, Department of Geography, Zürich, Switzerland; [2]University of Zurich, Department of Chemistry, Zurich, Switzerland; [3]University of Zurich, Department of Evolutionary Biology and Environmental Studies, Zurich, Switzerland; [4]International Centre of Insect Physiology and Ecology (ICIPE), Plant Health, Nairobi, Kenya; [5]University of the Free State, Department of Zoology and Entomology, Bloemfontein, South Africa; [6]South Eastern Kenya University, Department of Life Sciences, Kitui, Kenya; [7]Université Paris-Saclay, INRAE, AgroParisTech, UMR EcoSys, Palaiseau, France

*For correspondence:
daria.odermatt@bluewin.ch
(DMO);
meredithchristine.schuman@uzh.ch (MCS)

## eLife Assessment

Research on push-pull systems has often focused on controlled environments, leaving significant gaps in our understanding of how these systems function under real-world conditions. This **important** and **solid** study makes a substantial contribution by investigating the volatile emissions and behavioral effects of Desmodium in natural and semi-field contexts which offer insights of broad interest for sustainable agriculture and pest management. While the authors rightly acknowledge some remaining limitations, the revised manuscript now provides a well-supported and transparent assessment of the ecological role of Desmodium volatiles in push-pull systems.

**Abstract** Push-pull systems for sustainable pest management combine repellent stimuli from intercrops ('push') and attractive stimuli from border plants ('pull') to repel herbivorous insects from a main crop and attract the herbivores' natural enemies. The most widespread implementation, intercropping the legume *Desmodium* with maize surrounded by border grass, reduces damage from the invasive fall armyworm (FAW) *Spodoptera frugiperda*. While initial research indicated that *Desmodium* volatiles can dampen the attraction of FAW to maize, a recent study recovered very low volatile emission from the commonly used *D. intortum* and found that the *D. intortum* headspace did not reduce FAW oviposition on maize (Erdei et al., 2024). Here, we detect volatiles from the headspace of two *Desmodium* species sampled within the activity window of FAW: *D. intortum* and the more recently adopted *D. incanum*; and we present the behavior of gravid FAW moths in bioassays. We detected 25 volatiles from field-grown *Desmodium*, many in the headspaces of both species, including volatiles previously reported to repel lepidopteran herbivores. In cage oviposition assays, FAW moths preferred to oviposit on maize over *Desmodium*, but not on maize further from, versus closer to *Desmodium* plants that were inaccessible to the moths, but sharing headspace. In flight tunnel assays, moths approached the headspace of maize more than shared headspaces of maize and *Desmodium*, but pairwise differences were often insignificant. Thus, headspaces of

*Desmodium* species include volatiles that could repel FAW moths, and gravid moths were generally more attracted to maize and its headspace than to either *Desmodium* species or mixed maize-*Desmodium* headspaces. However, our results suggest that direct effects of *Desmodium* volatiles on FAW behavior are insufficient to explain reduced FAW infestation of maize under push-pull cultivation.

## Introduction

The sustainable intensification of agriculture is essential to achieve Sustainable Development Goal (SDG) 2, zero hunger, and reduce hidden costs of meeting requirements of other SDGs such as 13, climate action (*FAO et al., 2020*). This is particularly important in the developing Global South where larger yield gaps are observed compared to countries with greater financial resources (*Tilman et al., 2011*). Intercropping may offer sustainable solutions with potential for strong positive effects on pest and disease control as well as associated biodiversity (*Beillouin et al., 2021*).

The concept of 'push-pull' mixed cropping was first reported in 1987 as a 'stimulo-deterrence' strategy for the reduction of pest damage in cotton, combining a pest repellent intercrop ('push') interspersed with the main crop, and an attractant border crop ('pull') to trap pests at the field perimeter (*Pyke et al., 1987*). As most widely practiced, current push-pull systems benefit from the attractive and repellent stimuli of perennial inter- and border crops (referred to as companion crops), thereby reducing or eliminating the need for pesticides (*Pickett et al., 2014*). In Kenya, the first push-pull system, introduced in 1997, was designed to reduce yield losses caused by stemborer species and comprised a combination of the main crop maize (*Zea mays* L), the intercrop molasses grass (*Melinis minutiflora*), and the border crop Napier grass (*Pennisetum purpureum*) or Sudan grass (*Sorghum vulgare sudanense*) (*Khan et al., 1997b*).

Several volatiles detected from molasses grass repelled female stem borers and simultaneously attracted their parasitoid *Cotesia sesamiae* (*Khan et al., 1997a*). Following the initial success of this system, there was interest in replacing molasses grass with a leguminous repellent intercrop that could improve soil quality by fixing nitrogen while providing high-quality fodder. The legumes silver-leaf *Desmodium* (*Desmodium uncinatum*) and its congeneric (*Desmodium intortum*) were shown to repel ovipositing stem borers and furthermore to suppress parasitic *Striga* weeds (*Khan et al., 2000*; *Khan et al., 2002*). The first *Desmodium*-intercropped version of push-pull used *D. uncinatum* in place of *M. minutiflora* together with *P. purpureum*. As a result of screening for more drought-tolerant companion plants, a version using *D. intortum* combined with the border crop *Brachiaria* cv Mulato II was developed as 'climate-adapted push-pull', which has been demonstrated to increase maize yield by a factor of 2.5 (*Midega et al., 2015*). Recently, a 'third generation' push-pull system was evaluated using the intercrop *D. incanum* to replace *D. intortum* – which does not flower and set seed in many parts of tropical Africa and thus limits adoption and spread of push-pull by limiting seed supply for the intercrop – with *Brachiaria brizantha* cv Xaraes as border crop, which is more resistant to herbivorous mites that attack the mulato cultivar (*Cheruiyot et al., 2021*).

The invasive fall armyworm (FAW), *Spodoptera frugiperda*, originating from a maize-specialized strain from northern and central America, has become a major threat to African maize crops since 2016 (*Goergen et al., 2016*; *Day et al., 2017*; *Hailu et al., 2021*; *Zhang et al., 2023*). It spread rapidly through East Africa, with most of the farmers in Ethiopia and Kenya encountering the FAW after the long rains in the first half of 2017 (*Kumela et al., 2018*). The invasiveness of the moth is fueled by its relatively short life cycle of about 4 weeks and the capability of adult females to lay hundreds of eggs (*Sparks, 1979*). Other traits favoring the FAW's spread in Africa and beyond are its strong flight capacity, lack of diapause, reported survival in diverse habitats, rapid development of resistance to insecticides and viruses, and polyphagous nature (*Wan et al., 2021*). Although the FAW has a wide host range of at least 353 species across 76 plant families (*Montezano et al., 2018*), the high financial and yield losses in maize due to this invasive FAW strain are particularly devastating (*Eschen et al., 2021*). The FAW is reported to outcompete resident stemborers (*Hailu et al., 2021*; *Sokame et al., 2021*; *Mutyambai et al., 2022*). The use of pesticides is a popular approach to control the FAW, posing risks to environmental and human health including acute pesticide-related illnesses, as many smallholder farmers do not use personal protective equipment while spraying (*Tambo et al., 2020*). Therefore, sole reliance on pesticides is not sufficient to manage FAW sustainably, and Integrated Pest

Management (IPM) strategies, such as the promotion of natural enemies, are desirable (*Nyamutukwa et al., 2022*; *Van den Berg and du Plessis, 2022*). Climate-adapted and third-generation push-pull systems have been reported to reduce plant damage and yield loss caused by the FAW (*Midega et al., 2018*; *Hailu et al., 2018*; *Cheruiyot et al., 2021*; *Yeboah et al., 2021*; *Mutyambai et al., 2022*).

A systematic review on the chemical ecology of push-pull systems by *Lang et al., 2022* found only 30 publications (7 primary sources and 23 publications reporting on results from these primary sources) on the chemistry of push-pull and related mixed-cropping systems, from which 206 substances were reported to be potentially associated with push and pull effects. Of these, 101 were categorized as plant volatiles and reported by studies with a focus on plant-insect interactions (as opposed to studies with a focus on *Striga* control). However, none of these publications reported volatiles sampled under field conditions. Two recent papers, both published after the literature review conducted by *Lang et al., 2022*, reported additional potentially bioactive substances from companion plants in maize-*Desmodium*-grass push-pull systems, and both publications supported the hypothesis that volatiles from the companion crop *D. intortum* repel the FAW and attract parasitic wasps (*Sobhy et al., 2022*; *Peter et al., 2023*). In contrast, a third recent publication by *Erdei et al., 2024* detected low levels of volatiles from *D. intortum* plants and found that the presence of *D. intortum* upwind from maize did not reduce oviposition on maize, while FAW larvae offered *D. intortum* fed on it and became

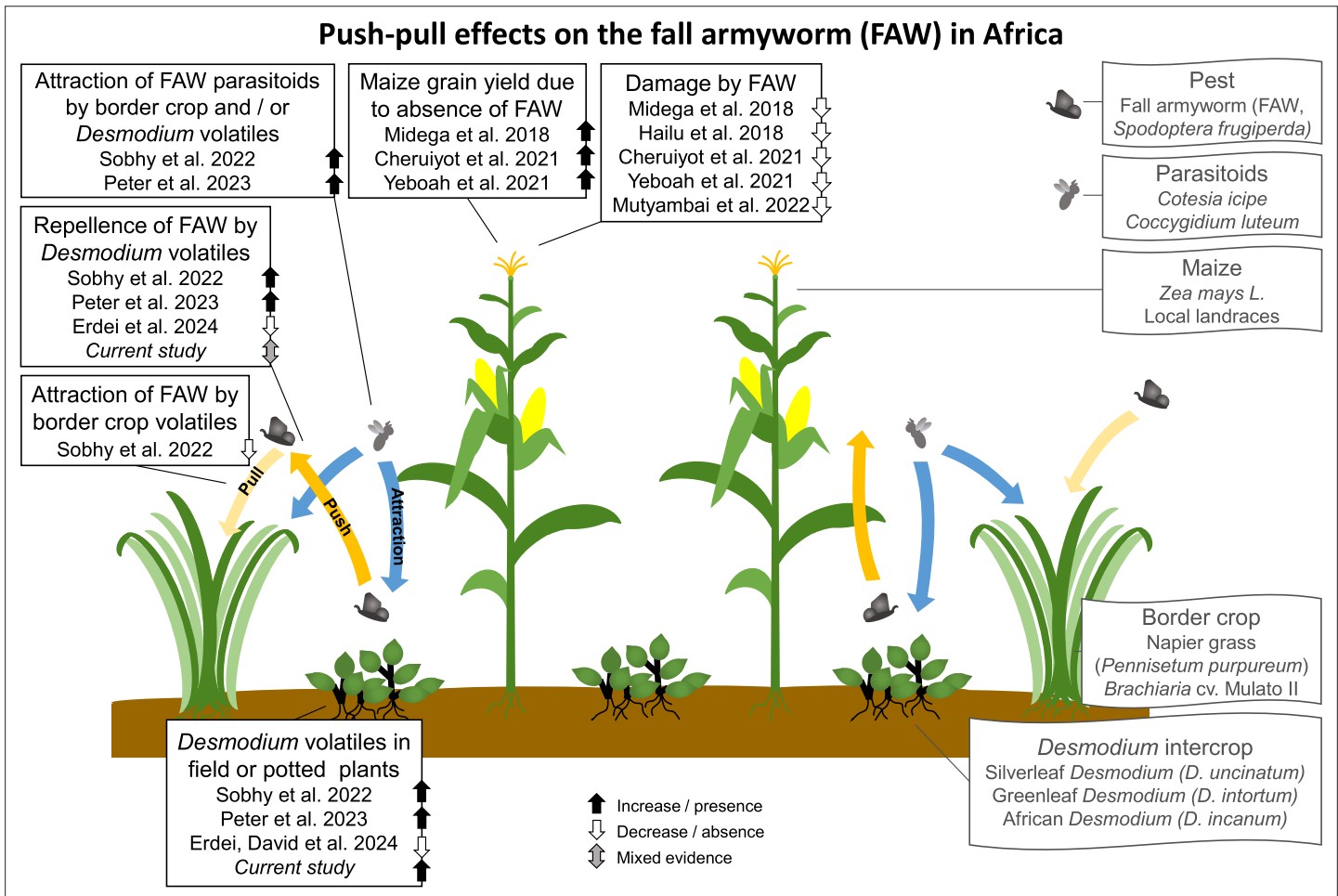

**Figure 1.** Studies to date on push-pull effects on the fall armyworm (FAW) embedded in the mechanisms according to the current state of knowledge of the system. Volatiles (and potentially other traits) of the intercrop repel the herbivorous insect and additionally attract its parasitoids, while volatiles (and potentially other traits) of the border crop attract herbivores away from the main crop (*Pickett et al., 2014*; *Khan et al., 2018*; *Eigenbrode et al., 2016*). The pull effect of the border crop Napier grass, *Pennisetum purpureum,* was observed in earlier systems, as it attracted stemborers (*Khan et al., 1997b*), but could not be confirmed for the FAW with the border crop *Brachiaria* cv Mulato II (*Sobhy et al., 2022*). For more detail on all studies, see *Supplementary file 1*.

**Table 1.** Mass fragments of the unknown features detected by EI-MS in field *D. intortum* and / or *D. incanum*.

| Name | Retention index | Peaks with relative intensity >5% [m/z (height in %)] |
|---|---|---|
| Monoterpene | 1229 | 93 (100.0), 121 (29.4), 77 (28.2), 120 (11.6), 105 (10.0), 80 (10.0), 136 (8.8), 107 (8.0), 91 (6.7), 92 (5.7) |
| Benzene derivative | 1356 | 119 (100.0), 117 (54.6), 134 (30.8), 91 (18.9), 118 (18.7), 115 (17.9), 102 (12.4), 120 (8.6), 77 (7.2), 65 (6.0), 51 (5.5), 63 (5.1) |
| Sesquiterpene1 | 1477 | 119 (100.0), 105 (94.6), 161 (77.1), 93 (45.7), 91 (39.5), 81 (26.8), 92 (26.3), 120 (23.7), 41 (19.1), 77 (17.7), 79 (15.8), 133 (14.3), 121 (14.0), 204 (13.6), 106 (13.5), 107 (13.3), 55 (12.3), 117 (12.3), 162 (11.1), 43 (10.0), 136 (8.4), 69 (7.3), 115 (7.2), 65 (6.6), 53 (6.4), 94 (6.4), 131 (6.0), 39 (5.8), 67 (5.7) |
| Sesquiterpene2 | 1716 | 105 (100.0), 93 (77.0), 119 (49.5), 91 (39.9), 41 (39.1), 107 (32.9), 79 (32.5), 161 (30.8), 94 (30.6), 81 (29.3), 69 (28.4), 55 (24.7), 204 (24.2), 133 (23.2), 77 (22.8), 189 (18.1), 106 (17.4), 121 (17.4), 147 (13.1), 53 (13.1), 92 (12.7), 95 (11.3), 109 (10.5), 39 (10.1), 120 (8.9), 162 (8.5), 67 (8.5), 134 (8.3), 108 (8.3), 82 (8.1), 43 (7.8), 122 (7.6), 65 (7.1), 148 (6.9), 175 (6.6), 123 (6.6), 54 (6.3), 135 (6.0), 80 (6.0), 117 (5.6), 110 (5.3) |
| Sesquiterpene3 | 1762 | 81 (100.0), 93 (99.0), 107 (84.5), 91 (74.1), 79 (70.5), 80 (68.5), 55 (56.8), 105 (54.9), 43 (54.3), 95 (53.8), 119 (51.7), 108 (50.6), 41 (50.0), 94 (48.6), 109 (42.8), 67 (41.7), 133 (35.1), 77 (32.9), 70 (28.9), 121 (28.7), 147 (23.9), 189 (22.9), 53 (20.4), 106 (19.3), 59 (18.6), 122 (17.9), 39 (17.8), 123 (17.4), 92 (16.9), 57 (16.6), 134 (16.4), 65 (16.2), 71 (15.3), 161 (15.0), 120 (13.7), 56 (13.5), 175 (13.0), 82 (12.0), 113 (10.9), 78 (10.2), 204 (9.2), 148 (9.0), 117 (8.7), 42 (8.2), 135 (8.1), 149 (7.8), 83 (7.7), 66 (7.5), 68 (7.2), 85 (7.2), 132 (7.2), 126 (6.9), 124 (6.7), 162 (6.5), 96 (6.4), 69 (6.3), 131 (6.1), 97 (5.9), 63 (5.6), 110 (5.3), 111 (5.0), 136 (5.0) |
| Naphthalene derivative | 1834 | 142 (100.0), 141 (81.3), 115 (34.7), 91 (20.4), 143 (11.3), 43 (11.3), 139 (10.9), 105 (8.8), 135 (7.7), 71 (7.3), 63 (5.6), 147 (5.1) |

trapped by its silicate trichomes; the authors thus proposed that reduced FAW infestation in maize-*Desmodium* intercropping settings is more likely to result from physical trapping of FAW larvae by *Desmodium*.

The protection mechanisms of push-pull systems are assumed to be complex and to operate across multiple interactions, protecting maize not only from FAW but also from other pests and parasitic weeds, such as *Striga*. However, further research is needed to understand the mechanisms by which the maize-*Desmodium*-grass push-pull system can protect maize from FAW. A summary of the studies to date on push-pull effects on FAW can be found in *Figure 1* and *Supplementary file 1*. To our knowledge, no data on plant volatiles from the more drought-resistant *D. incanum* have been published so far. Therefore, this study focuses on the volatiles of *D. intortum* and *D. incanum*, aiming to capture the headspace of plants used by farmers under realistic conditions, rather than attributing the composition to specific causes such as damage. Additionally, our goal is to assess the effects of exposure to the headspaces of these plants, versus direct exposure to the plants, on the interaction of FAW with maize. We collected volatiles from plants in farmers' fields as well as in bioassay setups under semi-field conditions and conducted oviposition and flight tunnel bioassays to assess FAW moth preferences.

## Results
### *Desmodium* volatile profiles

A total of 25 substances were measured in at least 2/3 of field-collected samples from each of the *Desmodium* species, of which 11 occurred in both species. These substances include (*E*)-*β*-ocimene, (*Z*)–3-hexen-1-ol, (*Z*)–3-hexen-1-ol acetate, 1-octen-3-ol, 3-octanone, caryophyllene, (3*E*)–4,8-dimethyl-1,3,7-nonatriene (DMNT), germacrene D, indole, linalool, and (3*E*,7*E*)–4,8,12-Trimethyl-1,3,7,11-tridecatetraene (TMTT). Five peaks were not fully identifiable and were categorized based on mass spectra and relative retention times as a benzene derivative, a naphthalene derivative, a monoterpene, and two sesquiterpenes (for feature information, see *Table 1*). The sesquiterpene (*E*)-α-bergamotene was detected only in the headspace of *D. incanum*, while another unidentified sesquiterpene only occurred in *D. intortum* (see *Figure 2* for a heatmap with means, *Figure 3* for sample comparison, and *Figure 2—figure supplement 1* for a per-sample heatmap). Of these substances, (*Z*)–1,5-octadien-3-ol, 1,2,3-trimethylbenzene, 3-pentanol, and hexyl acetate are reported here for the first time in connection with the chemical ecology of push-pull cropping systems (for more detailed information on each volatile substance, see *Supplementary file 1*). Germacrene D and 1-hexanol were

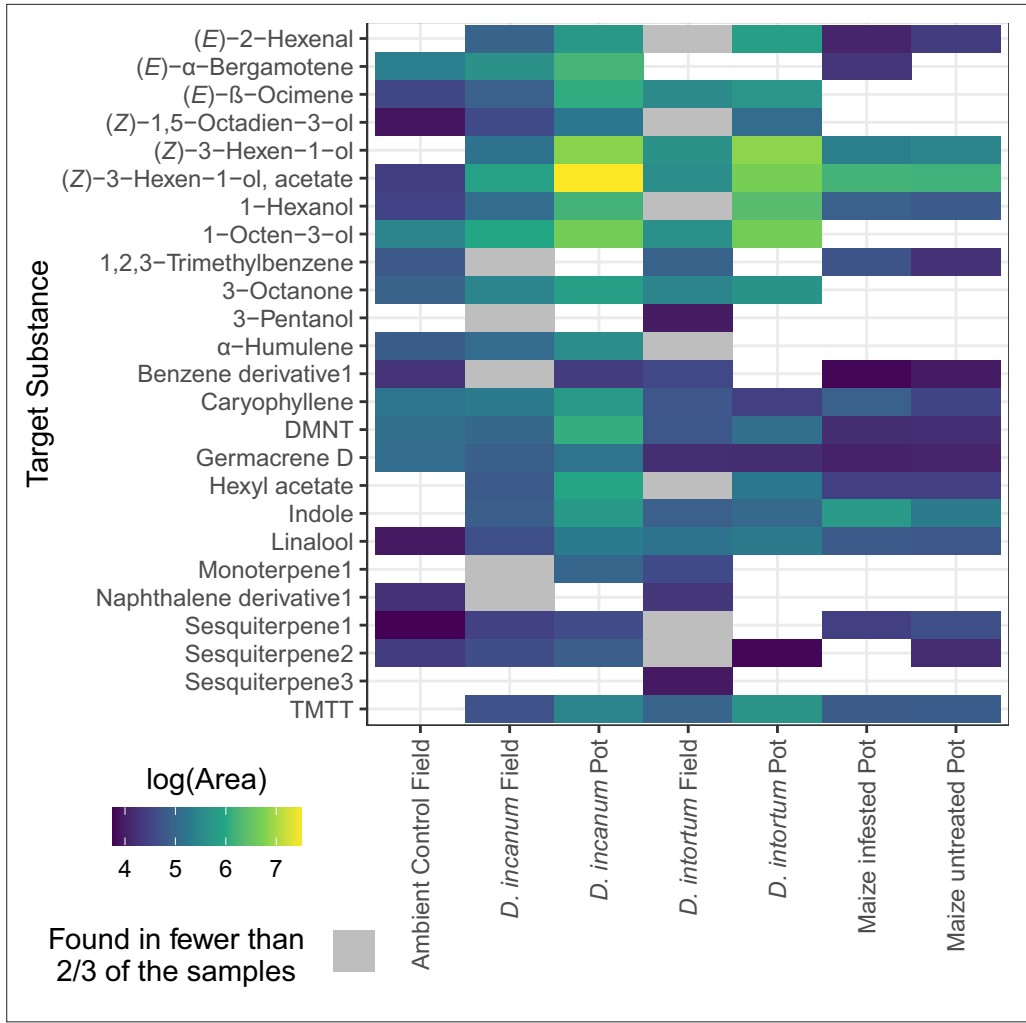

**Figure 2.** Heatmap comparing the log10-transformed peak area of the non-zero hits of substances present in at least 2/3 of the samples of at least one of the *Desmodium* species in the field, showing relative peak areas detected in field- or pot-grown (bioassay conditions) *D. intortum* and *D. incanum*, with pot-grown maize for comparison. Peak intensity can only be meaningfully compared within the same substance. The x-axis displays the species and their growth conditions (field or pot), and the y-axis shows the target substances in alphabetical order. The color indicates the mean of the log-10 transformed peak area of all samples with the substance present. The gray color indicates that a substance occurs in more than one, but less than 2/3 of the samples from the field. 'Maize Infested' refers to maize plants that were exposed to FAW eggs, as moths were allowed to oviposit on the plants two nights prior to volatile sampling. Sample sizes: Ambient Control Field = 4, *D. incanum* Field = 14, *D. incanum* Pot = 3, *D. intortum* Field = 11, *D. intortum* Pot = 5, Maize infested Pot = 4, Maize Pot = 4.

The online version of this article includes the following figure supplement(s) for figure 2:

**Figure supplement 1.** Heatmap showing the log10-transformed peak areas of target substances in species grown under field and pot conditions.

not yet reported from *Desmodium* species but were reported in relation to mixed cropping for pest control (*Lang et al., 2022*). Germacrene D was detected in elevated levels in maize plants grown in soil from push-pull fields in comparison with those grown in soil from maize monocultures (*Mutyambai et al., 2019*), while germacrene D and 1-hexanol both were reported to be emitted from bean plants, *Vicia faba*, and used by the black bean aphid, *Aphis fabae*, for host detection (*Pickett and Khan, 2016*; *Webster et al., 2008*).

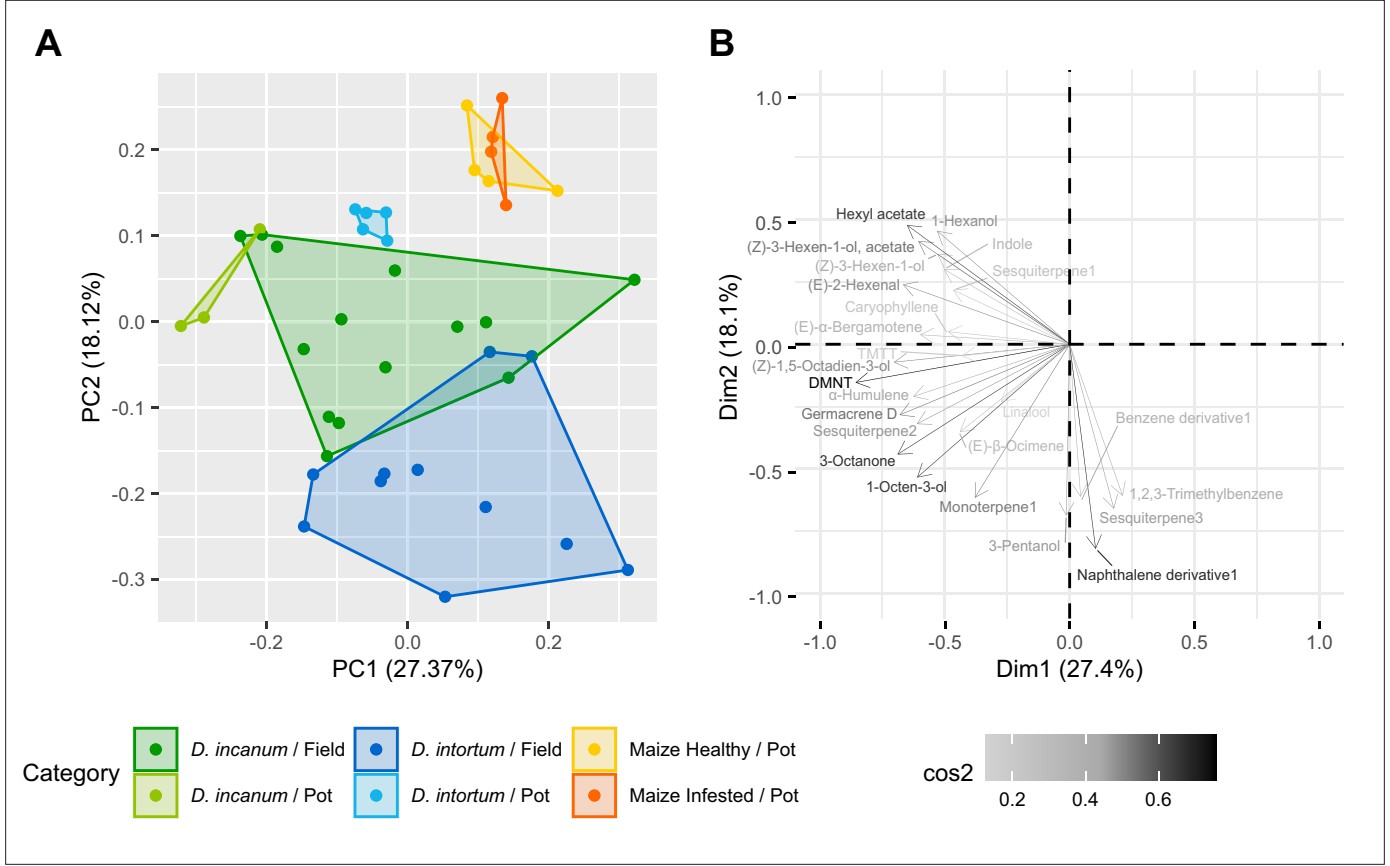

**Figure 3.** Principle component analysis of headspace volatile samples from field-grown and potted *Desmodium* spp. versus potted maize. (**A**) Principal component analysis plot based on the normalized peak areas of the 25 target substances. Sample sizes: Ambient Control Field = 4, *D. incanum* Field = 14, *D. incanum* Pot = 3, *D. intortum* Field = 11, *D. intortum* Pot = 5, Maize infested Pot = 4, Maize Pot = 4 (**B**) Loading plot of the projection of the variables on the first two dimensions. The x-axis displays PC1 (Dimension 1) and the y-axis shows PC2 (Dimension 2). The color indicates the cos2, whereby dark and long arrows indicate a better representation of a loading on these first two dimensions.

## Oviposition choice bioassays

Oviposition choice bioassays were conducted in cages to determine the influence of *Desmodium* plants and their headspace on the oviposition behavior of FAW moths. A fraction of the potted *Desmodium* plants used in the oviposition assays was sampled for volatiles the night before the start of trials, and the composition of these samples was compared with the substances and features detected from *Desmodium* plants sampled in farmers' fields. In the comparison between field *Desmodium* plants and potted bioassay plants that were sampled in a smaller number, the majority of substances showed a higher relative abundance in potted *D. incanum* (bioassay conditions), whereas in *D. intortum*, some substances detected in field samples occurred in higher concentrations in the potted plants, while others were not detected. Sixteen of the volatiles detected in *Desmodium* plants in farmers' fields were also present in either uninfested or infested potted maize plants (see *Figure 2*). In direct contact treatments, FAW moths were allowed to oviposit directly on *Desmodium* plants within the cage, and in treatments labeled as 'indirect', moths could choose between two maize plants, whereby one was closer to a *Desmodium* plant placed immediately outside the cage and thus closer to the headspace of that *Desmodium* plant (see *Figure 4*). Except for these 'indirect' treatments, plants were separated by at least 50 cm within and between cages, and positions were randomized (see Materials and methods).

The count of eggs and egg batches showed that fewer eggs were laid directly on *Desmodium* plants, with the egg count on maize plants being 7.9 or 6.8 times higher than on *D. incanum* or *D. intortum*, respectively, corresponding to a similar difference in the number of egg batches (*Figure 4*, *Figure 4—figure supplement 1*). A mixed model with egg counts on the maize plant as the observed

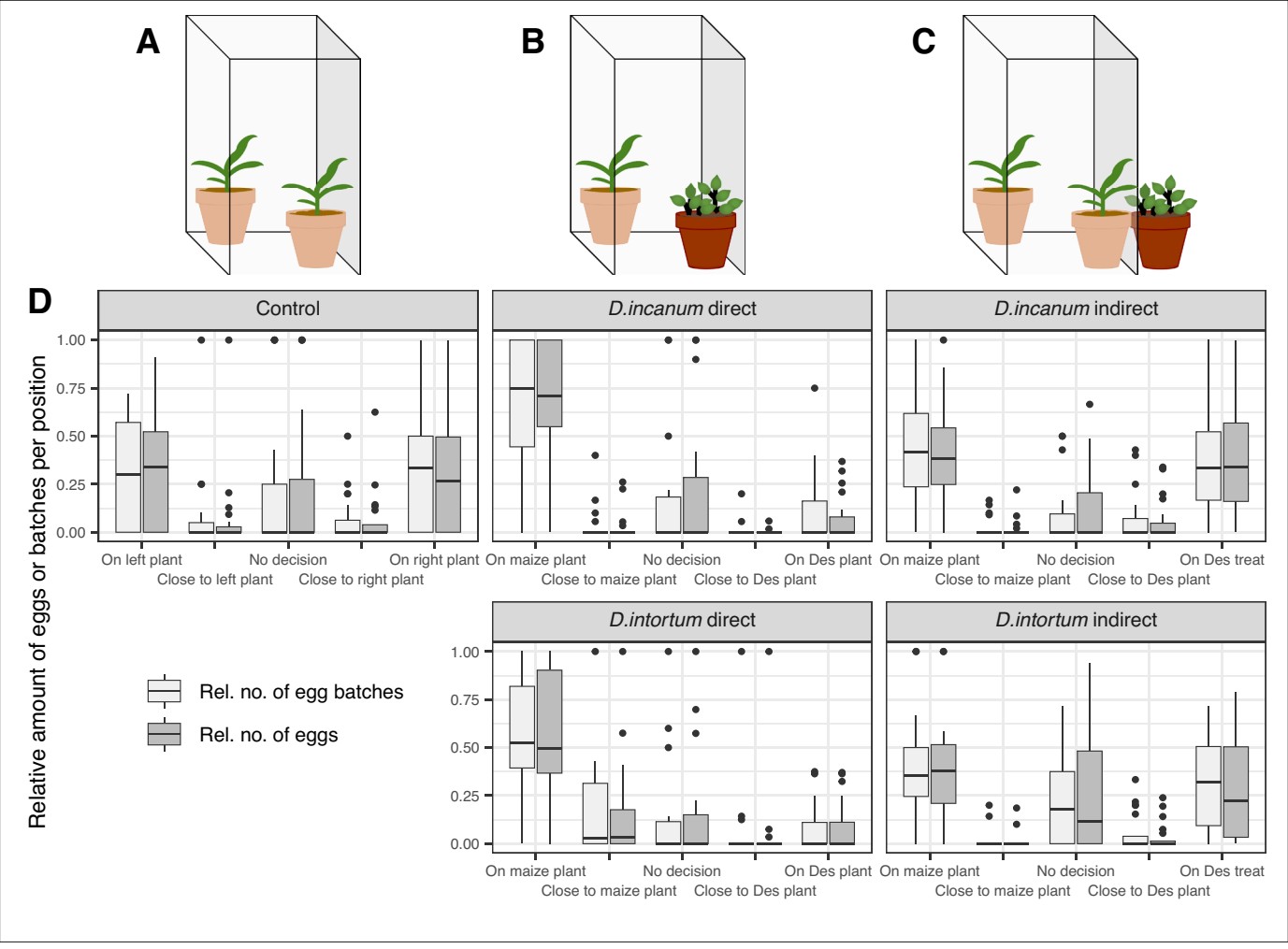

**Figure 4.** Results of choice oviposition assays for moths in cages. (**A**) Setup of the control with two maize plants inside the cage. (**B**) Setup of the treatments with direct contact with one maize and one *Desmodium* plant inside the cage. (**C**) Setup of the treatments with indirect contact with two maize plants placed inside the cage and one *Desmodium* plant placed in proximity to one of the maize plants, but unreachable for the moths. (**D**) The boxplots display the relative number of eggs (light gray boxes) or egg batches (dark gray boxes) per position (x axis) and treatment (superordinate boxes). The lower and upper hinges correspond to the 25th and 75th percentiles, while the whiskers extend to the largest and smallest non-outlier values, respectively. Outliers are values outside a window of 1.5 x the interquartile range and are plotted individually. *Desmodium* is abbreviated with 'Des' and treatment is abbreviated with 'treat'. See *Table 2* for a model-based statistical analysis and *Figure 4—figure supplement 1* for a breakdown of eggs per repetition and batch. Sample sizes (n, replicate units of a given treatment): Control = 21, *D. incanum* direct = 19, *D. intortum* direct = 20, *D. incanum* indirect = 19, *D. intortum* indirect = 20.

The online version of this article includes the following figure supplement(s) for figure 4:

**Figure supplement 1.** Graphs showing the total egg count per repetition and per batch during oviposition bioassays.

**Figure supplement 2.** Schemes and photographs showing the experimental setup of the choice oviposition bioassays.

**Figure supplement 3.** Scheme showing the cage setup in greenhouse 1B.

**Figure supplement 4.** Scheme showing the cage setup in greenhouse 3B.

**Figure supplement 5.** Photographic documentation showing the egg collection and counting process.

variable and accounting for plant position detected a significant difference between the control (two maize plants, no neighboring *Desmodium*) and all other treatments (F(1) = 11.11, p=0.0012), as well as for the comparison of direct versus indirect contact with *Desmodium* plants (F(1) = 9.51, p=0.0026). However, no significant difference was found between the two *Desmodium* species (F(1) = 1.08, p=0.3012; see *Table 2*).

**Table 2.** Mixed model applied for oviposition bioassays.

On maize plant: Comparison of the laid eggs on the maize plants for all treatments with the following formula: RelEggNoMaize ~ ContrvsTreat + IndvsDir + IncvsInt + Treatment + (1| Greenhouse) + (1| Group:CageNo) + (1| Start.Date) + (1| Rep).

On the *Desmodium* plant: Comparison of the laid eggs on the *Desmodium* plants for all treatments with the following formula: RelEggNoDesmodium ~ ContrvsTreat + IndvsDir + IncvsInt + Treatment + (1| Greenhouse) + (1| Group:CageNo) + (1| Start.Date) + (1| Rep).

Descriptions of the terms: RelEggNoMaize & RelEggNoDesmodium = The dependent variable was determined by number of eggs laid on maize or *Desmodium* relative to the total number of eggs laid in one repetition. ContrvsTreat = All *Desmodium* treatments are compared with the control treatment. IndvsDir = The treatments with direct and indirect contact with *Desmodium* plants are compared. IncvsInt = *D. incanum* and *D. intortum* are compared against each other. Treatment = All treatments are compared against each other to detect effects which are not related to terms used before. Greenhouse = Treatments were carried out equally in two different greenhouses. Group:CageNo = Twenty-five cages were repeatedly used, whereby each cage could be unambiguously identified with the group and cage number. Start.date = Treatments were carried out at five different start dates. Rep = All repetitions of the control were inserted twice with inversion of the values of the left and right maize plants. Any effects of the single repetitions are displayed in this term.

**On Maize plant**

| Term | Sum Sq | Mean Sq | Num DF | DenDF | F value | PR (>F) |
|---|---|---|---|---|---|---|
| ContrvsTreat | 1.0191 | 1.0191 | 1 | 111.09 | 11.1142 | 0.0012 |
| IndvsDir | 0.8722 | 0.8722 | 1 | 111.09 | 9.5117 | 0.0026 |
| IncvsInt | 0.0989 | 0.0989 | 1 | 112.10 | 1.0789 | 0.3012 |
| Treatment | 0.0634 | 0.0634 | 1 | 111.40 | 0.6915 | 0.4074 |

**On *Desmodium* plant**

| Term | Sum Sq | Mean Sq | Num DF | DenDF | F value | PR (>F) |
|---|---|---|---|---|---|---|
| ContrvsTreat | 0.3238 | 0.3238 | 1 | 111.27 | 5.3626 | 0.0224 |
| IndvsDir | 1.3604 | 1.3604 | 1 | 111.17 | 22.5287 | 6.20e-06 |
| IncvsInt | 0.0375 | 0.0375 | 1 | 112.61 | 0.6204 | 0.4325 |
| Treatment | 0.0599 | 0.0599 | 1 | 111.61 | 0.9927 | 0.3213 |

## Dual-choice assays

Dual-choice assays were conducted to compare short-term effects of plant volatiles on the flight behavior of FAW moths. The setup permitted only low air flow relative to its total volume, and so the possible mixing of stimuli both across trials and within a trial is not well defined and is likely more complex than in most laboratory trials and more similar to field and other less controlled settings. Overpressure was used to ensure the simultaneous transfer of headspaces from both treatments at equal airflow rates and from opposite directions into the setup, and moth movement toward these opposing sources was interpreted as choice. See the description of the methods and materials as well as the no-choice setup, which addresses limitations of this choice setup.

Two-thirds of the moths tested showed ten or fewer segment changes during the 5 min of each experiment. Across all treatments, at least 77% of the moths showed no movement in the last 2 min of the experiment or only little activity, with a maximum of two segment changes in the last 3 min of the experiment, which was interpreted to mean that they had made a decision within the 5-min time frame of the experiment. Overall, moths showed a reduction in segment changes over time, particularly for the comparison of maize vs. maize + *D. intortum* (see *Figure 5*). Across all treatments, a few individuals displayed high flight activity of up to 76 segment changes with no indication of decline over time.

No significant effects were detected when comparing the stays in the maize segments versus the *Desmodium* segments across the treatments using a mixed model (see *Table 3*). However, when comparing maize alone to maize with *D. intortum*, FAW moths tended to prefer the *D. intortum* treatment, with 2.5 times more individuals settling in that area and spending, on average, twice as

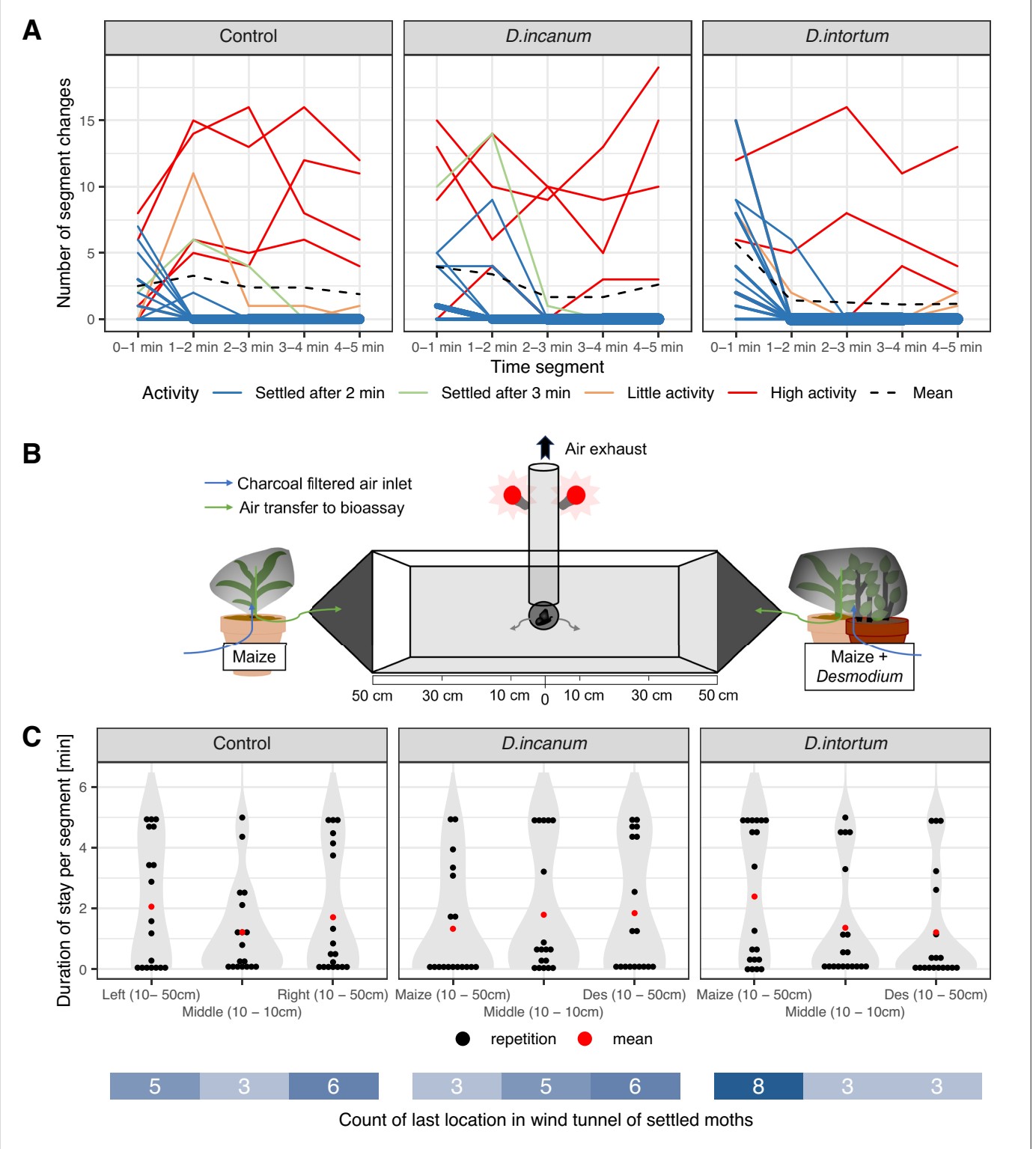

**Figure 5.** Results of dual-choice movement assays with plant headspaces. (**A**) Line plot displaying the number of segment changes per minute until the experiment ended after 5 min. The line width indicates the number of overlaying lines and the colors indicate the moth activity level: 'Settled after 2 min'=No segment changes after 2 min. 'Settled after 3 min'=No segment changes after 3 min. 'Little Activity'=Moths showed max. 2 segment changes in the last 3 min of the experiment. 'High Activity'=Moths showed higher activity with at least 6 segment changes in the last 3 min of the experiment. The mean of all repetitions is represented by the black dashed line. (**B**) Setup of the dual-choice assay for a *Desmodium* treatment comparing a maize plant (left) vs. a maize and a *Desmodium* plant (right). The moths are released through an opening in the center (at 0 cm) and

*Figure 5 continued on next page*

*Figure 5 continued*

observed for 5 min. (**C**) Violin plot displaying the duration of stays in each segment in treatments with *D. incanum* and *D. intortum* (comparison of maize and maize + *Desmodium*) with the inclusion of the individual data points per repetition (black) and their mean (red): The x-axis displays the location in the dual-choice assay and the y-axis the total duration that each moth stayed in each segment. Below the x-axis, the last position of the moths that were considered settled (settled after <3 min or little activity) is displayed. See *Table 3* for the statistical analysis and *Figure 5—figure supplement 1* for the set-up and segmentation of the dual-choice assay setup. Sample sizes (n, replicate units of a given treatment: Control = 18, *D. incanum* = 18, *D. intortum* = 19).

The online version of this article includes the following figure supplement(s) for figure 5:

**Figure supplement 1.** Photograph showing the experimental setup of the dual-choice assay with a *D. incanum* treatment on the left and maize on the right.

long there. In the comparison between maize and maize with *D. incanum*, only minor differences were observed in the mean residence time (less than 1 min); however, twice as many moths settled on the *D. incanum* side (see *Figure 5*). Fourteen of 19 moths exposed to airflow from maize vs. maize + *D. intortum* settled in a segment of the dual-choice assay within the first 3 min of the experiment, a larger proportion than in the maize vs. maize or the maize vs. maize + *D. incanum* treatment.

## No-choice assays

No-choice assays were conducted to assess the short-term effects of plant volatiles on the flight behavior of FAW moths in a setup with higher airflow and conducted in a blocked manner so that no mixing could occur between treatments (see Materials and methods). Treatments included a no-plant control as well as maize or maize + *Desmodium* for both *Desmodium* species. Repetitions in which moths exhibited movement in the final 2 min of the trial were considered non-decisive and excluded.

The initial distance, representing the first movement bigger than 5 cm made by a moth, did not show significant differences between treatments. However, the highest mean initial distance was observed in the maize treatment, while the maize + *D. intortum* treatment exhibited the lowest mean. The landing distance, defined as the final position relative to the release point, showed more pronounced treatment effects. Specifically, the maize treatment had the highest mean landing distance, while the

**Table 3.** Mixed model applied for dual-choice assay.

Side of Maize: Comparison of the duration of stay in the compartment of the maize plants for all treatments with the following formula: y_logitMaize ~ (MaizevsDes + Treatment) + PosMaize + (1|Date) + (1|Rep).

Side of *Desmodium*: Comparison of the duration of stay in the compartment of the *Desmodium* treatment for all treatments with the following formula: y_logitDesmodium ~ (MaizevsDes + Treatment) + PosMaize + (1|Date) + (1|Rep).

Descriptions of the terms: y_logitMaize and y_logitDesmodium = The dependent variable was determined by the logit-transformed accumulated duration of the stays in minutes in the two compartments (10 cm - 50 cm) closer to the maize plant or a maize plant combined with a *Desmodium* plant. ContrvsTreat = The *D. incanum* and *D. intortum* treatments are both compared with the control treatment. Treatment = The two *Desmodium* species are compared with each other. PosMaize = The position of the maize plant and the maize + *Desmodium* treatment were equally placed on the left or right side of the dual-choice assay. Rep = All repetitions of the control were inserted twice with inversion of the values of the left and right maize plants.

**Side of Maize**

| Term | Sum Sq | Mean Sq | Num DF | DenDF | F value | PR (>F) |
|------|--------|---------|--------|-------|---------|---------|
| ContrvsTreat | 3.2055 | 3.2055 | 1 | 67 | 0.2410 | 0.6251 |
| Treatment | 16.8770 | 16.8770 | 1 | 67 | 1.2687 | 0.2640 |
| PosMaize | 0.4577 | 0.4577 | 1 | 67 | 0.0344 | 0.8534 |

**Side of *Desmodium***

| Term | Sum Sq | Mean Sq | Num DF | DenDF | F value | PR (>F) |
|------|--------|---------|--------|-------|---------|---------|
| ContrvsTreat | 0.8454 | 0.8454 | 1 | 66.777 | 0.0618 | 0.8044 |
| Treatment | 13.2919 | 13.2919 | 1 | 64.002 | 0.9719 | 0.3279 |
| PosMaize | 6.5060 | 6.5060 | 1 | 48.989 | 0.4757 | 0.4936 |

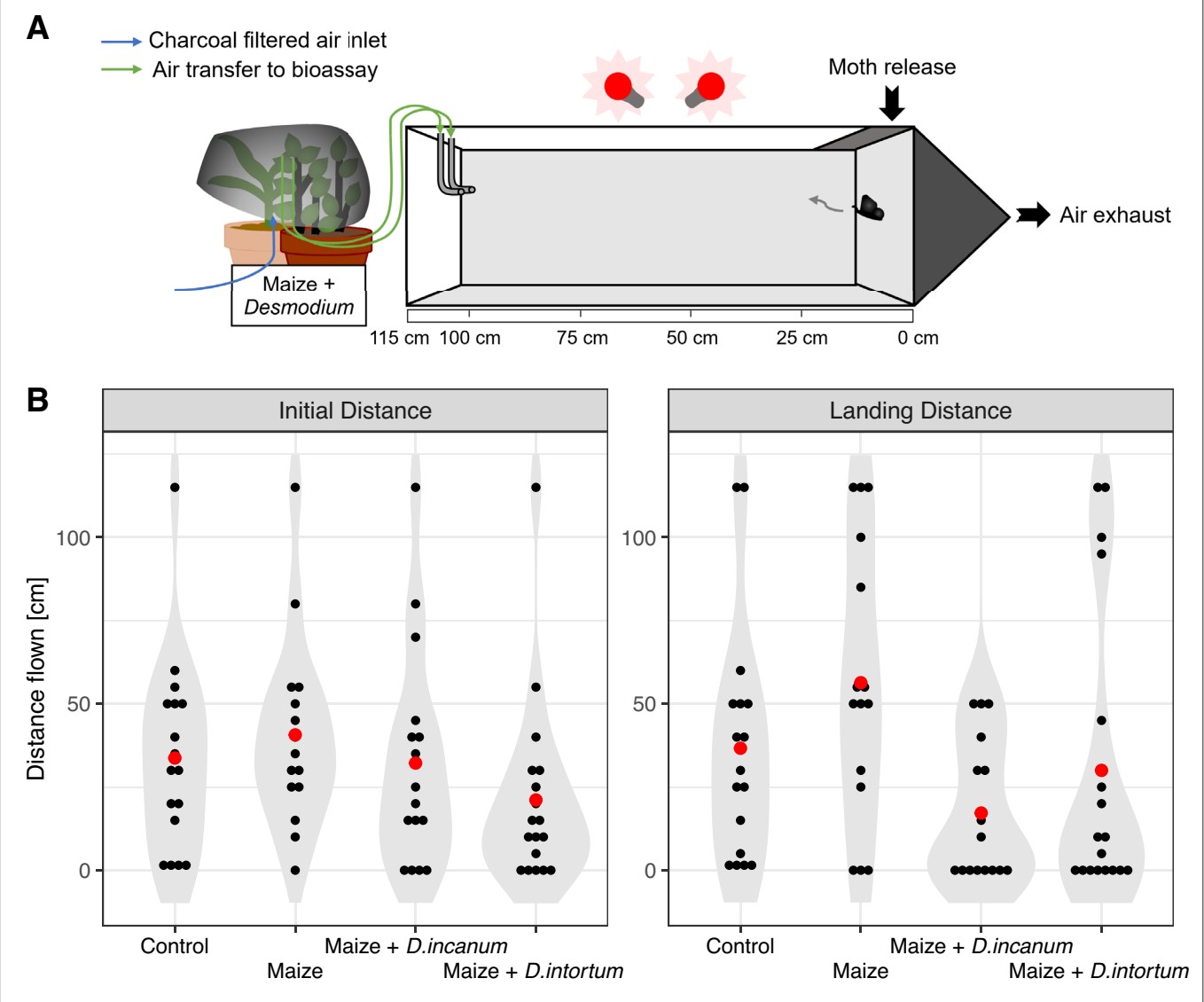

**Figure 6.** Results of no-choice movement assay with headspaces of maize or maize + *Desmodium*. (**A**) Setup of the no-choice assay for a *Desmodium* treatment. The moths are released through an opening in the right side (at 0 cm), and their movement is observed for 5 min. The plant odor was released from the outlet at 100 cm, while the moths were allowed to move freely up to 115 cm. All repetitions that showed activity in the last 2 min were excluded, as they are considered to have made no clear decision. (**B**) Violin plot with the inclusion of the individual data points per repetition displaying the distance flown in the first, initial movement and the final location that was considered the finally chosen position. The x-axis displays the treatment and the y-axis the total distance that a moth moved or flew. See **Table 4** for the statistical analysis. The sample sizes corresponded to 19–20 for all samples, whereby all repetitions with activity in the last 2 min of the experiment were removed. Remaining sample sizes (n, replicate units of a given treatment): Control = 17, Maize = 17, Maize + *D. incanum* = 16, Maize + *D. intortum* = 17.

maize + *D. incanum* treatment had the lowest (see **Figure 6**). Statistical analysis revealed a significant difference between the maize and the *Desmodium* treatments (see **Table 4**).

## Discussion

The fall armyworm (FAW) *Spodoptera frugiperda* invaded sub-Saharan Africa in 2016 and is responsible for major yield losses from maize crops (**Day et al., 2017**; **Goergen et al., 2016**). Push-pull cropping of maize with *Desmodium* and border grasses is reported to reduce FAW damage (**Midega et al., 2015**; **Cheruiyot et al., 2021**; **Mutyambai et al., 2022**; **Yeboah et al., 2021**; **Hailu et al., 2018**) and to increase maize yield (**Midega et al., 2018**; **Cheruiyot et al., 2021**; **Yeboah et al., 2021**).

**Table 4.** Mixed model applied for no-choice bioassays.

Initial Distance: Comparison of the distance flown in the first movement bigger than 5 cm with the following formula: InitialDistance ~ ContrvsTreat + MaizevsDes + Treatment + (1|Date) + (1|RepNo).

Landing Distance: Comparison of the landing distance of the moths that showed no movement for the last 2 min of the experiment with the following formula: LandingDistance ~ ContrvsTreat + MaizevsDes + Treatment + (1|Date) + (1|RepNo).

Descriptions of the terms: InitialDistance = Distance flown of a moth in the first movement bigger than 5 cm from the release point. LandingDistance = Final landing distance of a moth after the last movement. All repetitions with movements in the last 2 min of the experiment were considered as not decisive and excluded from the statistics. ContrvsTreat = All treatments including plants are compared to a control treatment with only an empty bag. MaizevsDes = All *Desmodium* treatments (maize + *Desmodium* plant) are compared with the maize (alone) treatment. Date = Date the experiment was conducted. RepNo = Order in which the repetitions were measured. Treatment = All treatments are compared against each other to detect effects which are not related to terms used before.

**Initial Distance**

| Term | Sum Sq | Mean Sq | Num DF | F value | PR (>F) |
|---|---|---|---|---|---|
| ContrvsTreat | 0.46 | 0.46 | 1 | 0.0005 | 0.9867 |
| MaizevsDes | 1161.3 | 1161.3 | 1 | 1.3287 | 0.4949 |
| Treatment | 155.7 | 155.7 | 1 | 0.1781 | 0.7716 |

**Landing Distance**

| Term | Sum Sq | Mean Sq | Num DF | F value | PR (>F) |
|---|---|---|---|---|---|
| ContrvsTreat | 11.1 | 11.1 | 1 | 0.0085 | 0.9270 |
| MaizevsDes | 12771.6 | 12771.6 | 1 | 9.7598 | 0.0028 |
| Treatment | 1450.2 | 1450.2 | 1 | 1.1082 | 0.2976 |

Push-pull cropping systems benefit from the attractive and repellent stimuli of perennial companion crops (*Pickett et al., 2014*), with more than one hundred plant volatiles from different classes, especially terpenoids and fatty acid-derived 'green leaf' volatiles, reported to be associated with push-pull effects (*Lang et al., 2022*). Nevertheless, the repellent role of volatiles from greenleaf *Desmodium* (*D. intortum*) has been investigated in only three studies, with conflicting reports of whether these volatiles repel FAW moths, and the only study reporting field measurements recovering few volatiles in relatively low abundance (*Sobhy et al., 2022*; *Peter et al., 2023*; *Erdei et al., 2024*).

We characterized volatiles sampled from both *D. intortum* and a recently introduced alternative *Desmodium* intercrop, *D. incanum*, growing in farmer fields or in bioassay conditions, and conducted bioassays with FAW moths to determine whether the presence or proximity of *Desmodium* plants or their headspace would alter moth preferences for maize. We identified up to 17–19 substances in headspace samples from *Desmodium* plants, often in abundance similar to or greater than volatiles measured from maize plants under comparable conditions. We observed that FAW moths preferred to oviposit on maize rather than on *Desmodium* plants of either species, but that close proximity to a *Desmodium* plant sharing the airspace did not influence FAW oviposition preferences for maize in a greenhouse cage choice assay. Consistent with this, in no-choice assays, FAW flying upwind landed closer to a maize odor source than when maize was combined with either species of *Desmodium*, but in dual-choice assays, FAW moths did not stay significantly closer to maize headspace sources over headspace sources from a combination of maize and *Desmodium* plants. We note that the choice assays (oviposition and dual-choice movement) were performed in relatively complex odor backgrounds, whereas the no-choice assay was performed with a simpler odor environment.

Furthermore, all assays were performed in a risk area for malaria and other mosquito-borne illnesses, so experimenters were protected with DEET, which has low volatility but is active in the vapor phase, where it can cause host avoidance by binding to insect olfactory receptors (although this has been mostly studied in the dipterans *Aedes aegypti* and *Drosophila melanogaster*) (*DeGennaro, 2015*). We reduced any potential influence on the lepidopteran in our experiments by using minimal amounts of DEET, avoiding direct contact of any DEET with experimental materials, and conducting behavioral assays without the insects being exposed to volatiles from the experimenter (experimenters were

generally not present during oviposition assays, and only small amounts of DEET were used during the movement assays indoors in Nairobi).

We conclude that *Desmodium* plants in push-pull cultivation emit volatiles that have the potential to repel FAW (and to attract their natural enemies), but our bioassays provide only weak support for the hypothesis that these volatiles repel FAW from maize. While (perhaps subtler) effects of the *Desmodium* headspace might be more reliably detected using more refined experimental setups, our mixed support of this hypothesis is in line with the reports in literature thus far (*Figure 1*). We suggest that other features of the maize-*Desmodium*-border grass push-pull system, including the influence of border grasses as well as below-ground interactions and other traits of *Desmodium*, are important for protecting maize from FAW moths and their larvae. We furthermore call for field or semi-field bioassays that better capture the choice of moths exposed to push-pull versus non-push-pull fields, in contrast to individual headspace environments presented out of context, or maize plants closer to vs. further from *Desmodium* plants in pots.

## Emission of volatiles implicated in push-pull by *Desmodium* intercrops

The headspace of *Desmodium* plants was dynamically sampled on Tenax TA adsorbent directly in push-pull fields, as well as from potted plants used for oviposition bioassays, overnight. Due to limited resources and time, we chose to focus on the volatile profiles of field-grown *Desmodium* rather than field-grown maize or intercropped edible legumes, as this aligned best with our aim of assessing the role of *Desmodium* volatiles. Additionally, we opted not to use individual volatile substances in bioassays because the conflicting evidence in the literature made it difficult to select meaningful blends, and testing plant emissions first was crucial for ensuring ecological relevance in our experiments. However, we recommend that future studies sample the volatiles of all plants commonly used in newly developed push-pull systems, including the edible legumes that are now more often intercropped, allowing for a contextualized comparison of volatile profiles.

We excluded all substances found in fewer than two-thirds of the field samples from further analysis. While this threshold is arbitrary and may overlook potential differences in plant stressors across fields, we chose to focus on substances commonly present in field-grown *Desmodium* plants. Quantitative comparisons within single substances are possible, but must be handled with care, as normalization based on the biomass and leaf area of the plants was not possible. The chosen adsorbent, Tenax TA, is known to be effective for lipophilic to medium-polarity organic compounds of intermediate molecular weight (approximately C7-C26) (*Dettmer and Engewald, 2002*; *Tholl et al., 2020*). It is possible that headspace samples changed during storage (up to 3 months tightly closed at room temperature) as degradation or thermal rearrangements on Tenax TA material have been reported (*Dettmer and Engewald, 2002*; *Alborn et al., 2021*). We extracted headspace samples using thermal desorption, which is simple and sensitive (*Tholl et al., 2006*), but not appropriate for some substances, such as the sesquiterpene germacrene A, which are susceptible to thermal rearrangement or degradation (*Faraldos et al., 2007*). While the potted plants were kept in insect-proof greenhouses, it cannot be excluded that field plants might have been exposed to insect herbivory, though representative and healthy plants were selected for the volatile sampling. Although plants were handled carefully, it is still possible that volatile release could have resulted from physical damage.

Despite these caveats, our samples yielded a variety of volatiles with high signal:noise ratios, of which 13 had been previously reported from sampling the headspace of potted *D. intortum* plants onto Porapak Q for sampling durations of 24 hr or 48 hr followed by solvent extraction and GC-MS analysis (*Sobhy et al., 2022*; *Peter et al., 2023*). In contrast, *Erdei et al., 2024* detected fewer volatiles in lower relative abundance from *D. intortum* using solid-phase microextraction (SPME) and a saturation time of 18–24 hr to sample intact, mechanically damaged, or herbivory-induced potted plants or intact plants in Tanzanian and Ugandan fields. Our study is the only one of these that limited the window of sampling strictly to the nighttime hours corresponding to the reported activity window for the FAW (*Sparks, 1979*). In pretests, no shorter activity window could be determined as the moths still showed mating behavior and potential oviposition at around midnight (see Appendix 1). Differences may be explained in part by different sampling techniques, as the sensitivity of SPME can be lower in comparison to dynamic headspace collection (*Tholl et al., 2006*). To our knowledge, the volatile profile of *D. incanum* was not previously reported. In conclusion, our data are consistent with the hypothesis that *D. intortum* and *D. incanum* emit volatiles, many of which have been previously

associated with push-pull effects, the repellence of lepidopteran herbivores or the attraction of their natural enemies.

## Oviposition choice bioassays

FAW oviposition behavior was observed over three nights in two treatments per *Desmodium* species. In one treatment, FAW moths had direct contact with *Desmodium* plants, and in another, they had only indirect contact through aerial exposure to plants outside of, but directly next to a mesh cage. Moths always had direct contact with maize plants.

Critical consideration must be given to differences in the volatile profiles between the potted plants and those measured in farmers' fields, which led to a separation of potted and field *D. intortum* in the second dimension of a PCA, and a near-separation of *D. incanum* in the first dimension. As the untargeted analysis was conducted based on the field *Desmodium* samples, any volatile substances that only occur in the potted plants were not identified or analyzed further (although full results of the GC-MS analysis are available with the source data for this publication). Oviposition on maize can alter the maize volatile profile and affect volatile emission in the following days, which is why the maize samples are divided into plants before (control) and after (infested) oviposition exposure. Of the substances occurring in field *Desmodium* plants, approximately half were abundant in healthy and infested potted maize plants, with only one substance, (*E*)-α-bergamotene, being present solely in infested maize plants. The sampling of potted plants was conducted using the same procedure as in the field, but in a greenhouse with netted walls on two sides that allowed air circulation, as used for the oviposition bioassays. Overall, this highlights the importance of studying oviposition cues and moth preferences directly in push-pull fields in the future.

In all treatments, the number of eggs laid per batch showed no difference depending on the position, which indicates that FAW moths did not adjust their behavior once they started laying eggs. This differs slightly from the outcomes reported by *Peter et al., 2023*, where a reduction in egg number per batch was observed and the number of eggs, but not the number of egg batches, differed between maize vs. maize + *D. intortum*. However, a clear preference for maize compared to the two *Desmodium* species was observed in the treatments where FAW moths had direct contact with both plant species, with a large effect size of 7–8 times more eggs (as measured by number of eggs or number of batches) on maize than on *Desmodium*. The preference of maize over *D. intortum* in oviposition assays with direct contact to both plants is consistent with various publications using similar setups (*Sobhy et al., 2022*; *Peter et al., 2023*; *Erdei et al., 2024*). There was little oviposition on the cage in our bioassays, which indicates that moths primarily chose between plants in our setup. This differs from the strong preference found by *Sobhy et al., 2022* of FAW moths to oviposit directly on the mesh of a bioassay cage rather than on the plants when offered maize co-planted with *D. intortum*. *Desmodium* and maize plants were planted in the same pots by *Sobhy et al., 2022*, and therefore maize volatiles may have been affected by belowground interactions or through priming by *Desmodium* volatiles prior to or during oviposition bioassays. This is, in fact, closer to the configuration in push-pull fields but does not isolate the influence of the *Desmodium* headspace, as we sought to do here.

prior to or during oviposition bioassays. This is, in fact, closer to the configuration in push-pull fields but does not isolate the influence of the *Desmodium* headspace, as we sought to do here.

In our bioassays, no significant effect could be seen in the treatments where moths had only indirect contact with *Desmodium* volatiles coming from a *Desmodium* plant proximate to one maize plant, but outside of the cage. This is inconsistent with the hypothesis that *Desmodium* volatiles repel FAW moths as proposed (*Pickett et al., 2014*; *Sobhy et al., 2022*). It is possible that the distance of about 60 cm between the plants in the cages was not suitable to create a gradient of volatiles sufficient to allow moths to choose between maize plants closer to versus further from *Desmodium*, in which case neighboring treatments with distances of 50–100 cm could have also affected each other. Although the distance reflected the approximate spacing between *Desmodium* and maize plants in push-pull fields, this limitation must be considered, as the close proximity of assays may have compromised the ability to create distinct volatile environments. To our knowledge, the only comparable setup was reported by *Erdei et al., 2024*, in which two maize plants were exposed to volatiles of either *D. intortum* or an artificial plant in a modified dual-choice assay, and moths did not show a significant preference for either maize plant.

In conclusion, we confirm a preference of FAW moths to oviposit on maize rather than on *D. intortum* or *D. incanum* when directly offered plants. However, no significant influence of the *Desmodium* headspace was observed. It is not to be expected that herbivores use all potential host plants equally, and preferences among host plants or for host over non-host plants can be based on different plant traits. Ovipositing FAW moths may also prefer maize over *Desmodium* due to other traits such as the surface texture, as proposed in the findings from *Erdei et al., 2024* that silicate trichomes on *D. intortum* pose a danger to FAW larvae; or chemical repellents directly on the leaf surface. Belowground interactions or priming between *Desmodium* and maize plants co-planted in push-pull fields may also change moths' preference for maize, an effect which could not have been observed with our experimental design, but might explain inconsistencies with findings from *Sobhy et al., 2022*.

## Dual-choice and no-choice assays

The short-term flight behavior of FAW moths was tested in a dual-choice assay setup where headspaces of a maize plant vs. maize + *Desmodium* were compared against each other, and a no-choice assay with a single headspace source upwind of the moth. Flight behavior showed large variation among the individual moths, but three-quarters of the moths had settled after at least 3 min or showed few changes in the last 2 min of each experiment. Although flow rates were measured at the incoming tubing of the dual-choice setup and the exhaust was placed above the center (where the moth was also introduced), this experimental setup allowed no control over the exact flows in the flight arena and the individual flows were low compared to the total arena volume. Therefore, the exact mixing of the opposing treatments is unknown, and cross-contamination between treatments was also possible. Furthermore, moths may behave differently over longer times or when given more space to maneuver, as under normal field conditions. The quarter of the moths that showed high activity indicates the difficulty of quantifying the responses of such active fliers. Interestingly, a larger proportion of moths settled after 2 min when presented with maize versus maize + *D. intortum* in the dual-choice assay, rather than continuing to fly. While some tendencies were apparent in this setup, none of the treatments showed any significant preference. In contrast, a significant repellent effect of *Desmodium* added to maize, versus maize alone, was observed in the no-choice assay as judged by final landing position. The combination of maize + *D. incanum* showed a greater repellent effect than maize + *D. intortum*. Similar repellence of moths by *D. intortum* headspace with or without the addition of maize was observed in no-choice assays reported in the literature when the total distance flown upwind at the end of the experiment was used as a metric (*Sobhy et al., 2022*; *Peter et al., 2023*).

## Conclusion

Several field studies have described positive effects of push-pull systems with the companion crops *D. intortum* or *D. incanum* on maize yield (*Midega et al., 2015*; *Midega et al., 2018*; *Cheruiyot et al., 2021*). Here, we show that 17–19 volatile substances, including plant volatiles previously indicated to repel lepidopteran herbivores or attract their natural enemies, were found in the headspaces of both companion crops *D. intortum* and *D. incanum* within the temporal activity window of FAW moths. However, we did not observe a repellent effect of *Desmodium* plant headspaces on FAW moths in

most bioassays. FAW moths clearly preferred to oviposit on maize over *D. intortum* or *D. incanum* plants. Our results thus indicate that moths prefer maize over *Desmodium* and that this may be influenced by short-range mechanisms such as an unfavorable leaf surface of *Desmodium*. However, the main interest in the context of push-pull is how *Desmodium* contributes to protecting co-planted maize. A reduction of FAW damage on maize under push-pull cultivation has been demonstrated in multiple studies (*Midega et al., 2018*; *Hailu et al., 2018*; *Cheruiyot et al., 2021*; *Yeboah et al., 2021*; *Mutyambai et al., 2022*). Based on the literature, effects of border crop volatiles may be important for reducing FAW damage in push-pull fields, in combination with the attraction of parasitoids by *Desmodium* volatiles and unfavorable host qualities of *Desmodium* plants for FAW (*Figure 1*). Furthermore, the *Desmodium* intercrop may be equally or more important for weed suppression, soil fertility, and improving maize plant health and vigor as for volatile-mediated repellence of FAW in the push-pull system. For example, recent studies showed that planting maize in soil collected from push-pull fields or co-planting it with silverleaf *Desmodium uncinatum* changed the composition of volatiles and defensive non-volatile substances in maize plants (*Mutyambai et al., 2024*; *Bass et al., 2024*). Future studies wishing to test the importance of *Desmodium* volatiles in the system could best resolve this question by adding and subtracting *Desmodium* headspace, or manipulating moths' direct access to *Desmodium* plants, in experimental push-pull and control fields, thus ensuring relevant context while avoiding experimental artifacts.

## Materials and methods

All volatile sampling and bioassays were performed with two *Desmodium* species, *Desmodium intortum* (greenleaf *Desmodium*) and *Desmodium incanum* (often referred to as African *Desmodium* by the push-pull farmers in Western Kenya), referred to in the methods section collectively as *Desmodium* for simplicity. All raw data and code used for the statistical analyses can be found on Zenodo (CERN, Geneva, Switzerland, https://doi.org/10.5281/zenodo.11633889) and GitHub (copy archived at *Odermatt, 2025*).

### Pretest moth activity window

The determination of the activity window of the FAW was relevant for the choice of the duration of the volatile sampling. The FAW is reported to be nocturnal (*Sparks, 1979*), without reports of a more precise activity window. For a first cycle, two single maize leaves of the landrace 'Jowi white' were placed in wooden cages with meshed side walls (100 x 60 x 60 cm) in high cylindrical glasses filled with water, but sealed with a piece of cotton to prevent moths from entering the glasses. Moths (3–4 days old) were released in pairs of one female and one male in 10 different cages (resulting in 20 moths in total) and observed from 7 pm to 11:30 pm over two consecutive nights. The moths were re-collected in falcon tubes in the morning after the first night, sealed with a piece of cotton to allow air circulation and stored at a temperature of approximately 25 °C until they were released in the same combinations on the following evening. For the second night, fresh maize leaves were provided for the moths to oviposit. For a second cycle, potted maize plants with multiple leaves of the landrace 'Jowi white' were placed inside the wooden cages with a piece of cotton soaked with water. Three female and two male moths of different ages were placed each in three cages (resulting in 15 moths in total) and observed from 6 pm - 1 am in two consecutive nights. During the daytime, the moths and the plant were left unaltered in the cage.

### Headspace sampling

We chose to focus on the volatile profiles of field-grown *Desmodium* rather than field-grown maize or intercropped edible legumes. This decision was driven by the relative scarcity of literature on the volatile profiles of *Desmodium* plants in real-world conditions of push-pull fields. To our knowledge, only one study has attempted to measure the volatiles of *D. intortum* *Erdei et al., 2024*, and no studies have measured volatiles of *D. incanum* in fields. Headspace samples of *Desmodium* were collected from plants in push-pull fields of local smallholder farmers near Mbita in the counties Homa Bay and Siaya, coordinates determined with WGS84 coordinate system (latitude, longitude): *D. incanum* = (–0.382096, 34.175487), (–0.4298279, 34.207891), (0.189957, 34.36072), *D. intortum* = (–0.382096, 34.175487), (–0.6302736, 34.494595), (–0.551479, 34.314673). *Desmodium* headspace samples

were collected in three different push-pull fields in May and June 2023 during the long rainy season, selecting four representative and healthy plants per field. As is becoming more common, some of the push-pull fields were mixed with vegetables such as kale or cowpeas. Headspace collections were also performed from potted *Desmodium* plants, as well as from healthy and infested potted maize plants on the research campus. Five 3-week-old maize plants, later labeled as 'infested', were exposed in two cages, each with three female and two male moths, for two nights prior to headspace sampling, which resulted in 1–10 batches per plant. Potted plants were placed in a greenhouse with a glass roof and two glass walls, in which air was able to circulate through netted walls on the opposing longer walls of the greenhouse.

As an adsorbent, commercially available Tenax TA glass tubes (containing 150 mg adsorbent, 35–60 mesh, Markes International Ltd, England or 186 mg adsorbent, 60–80 mesh, Merck, Germany) were used. A handful of leaves was enclosed in a roasting bag (Sainsbury's, London, UK), which was heated to 200 °C for at least 1 hr to clean the bag, then cooled to room temperature. The inlet tubing pushed 500 mL/min charcoal-filtered air into the bags from the lower bag rim, while the Tenax TA tube was connected to an air outlet tubing through a small hole at the top of the bag. Airflows were regulated by PYE Volatile Collection Kits (B. J. Pye, Kings Walden, UK) as described in *Steen et al., 2019*. Volatiles were accumulated on the adsorbent by aspirating the air at 200 mL/min (+/-40 mL/min) for 11.75 h (+/-30 min) overnight from 7 pm until 6.45 am on the following day. Under the same conditions, four ambient controls were sampled with an empty roasting bag on two fields with each *Desmodium* species. Additionally, a Tenax TA storage control was stored with all tubes, but never taken to the fields, and analyzed to check for contamination.

## TD-GC-MS measurement

All samples were measured on a TD-GC-MS instrument (Thermal Desorption – Gas Chromatography – Mass Spectrometry) from Shimadzu (TD30-QP2020NX, Kyoto, Japan). The Tenax TA tubes were desorbed for 15 min at 220 °C under 80 mL/min nitrogen flow. Volatiles were trapped on a Tenax cooling trap at –20 °C and after desorption was completed, the cooling trap was heated rapidly to 230 °C. The sample was injected with a split ratio of 1:5 and separated on a Rtx-Wax column (30 m, 0.25 mmID, 0.25 µm, Restek Corporation, USA) with the following 35 min oven program: holding at 40 °C for 2 min, heating to 150 °C with 10 °C/min, holding at 150 °C for 2 min, heating to 190 °C with 3 °C/min, heating to 230 °C with 10 °C/min and holding at 230 °C for 3 min. Mass fragments from 33 m/z to 400 m/z resulting from electron ionization at 70 eV were recorded with a scan rate of 5 Hz at an ion source temperature of 230 °C from 3 to 30 min.

## Feature detection

A feature list was generated in MZmine (Version 3.9.0, *Schmid et al., 2023*) from all substances that occurred in at least one-third of the field samples of either *Desmodium* species and in less than 4/5 of the combined control samples (ambient controls and storage control). Features were numbered and considered as unidentified if there was no corresponding reference substance. After removal of contaminants, the features were added to an existing target list that was composed with commercially available plant volatiles that can be mainly classified as terpenoids and benzenoids, which as a whole contain a large variety of functional groups such as alcohols, aldehydes, ketones, and esters. Additionally, the existing list included four green leaf volatiles ((*E*)–2-hexen-1-ol acetate, (*E*)–2-hexenal, (*Z*)–3-hexen-1-ol, (*Z*)–3-hexen-1-ol acetate) and a selection of more specific substances reported in push-pull fields such as the norsesquiterpene (*E*)–4,8–dimethyl–1,3,7-nonatriene (DMNT) and the norditerpene (*E*,*E*)–4,8,12-trimethyl-1,3,7,11-tridecatetraene (TMTT). The peak integration for all substances and features was manually checked for all control samples (ambient and storage control) using the LabSolutions Insight GCMS software (Shimadzu corporation, Kyoto, Japan). Peaks for all targets that occurred in at least 1/2 of the *Desmodium* field samples and in no more than 3/5 of controls were also manually checked for all samples. Finally, all targets that occurred in at least 2/3 of the field samples of *D. intortum* or *D. incanum* and in no more than 3/5 of the controls were considered as present. In *Table 1* the electron impact ionization (EI)-MS data of the present, unidentified features are displayed, and in *Table 5* the origin of the reference standards used to identify the target hits can be found.

**Table 5.** Origin of the reference standards of all identified target substances present in field *D. intortum* and / or *D. incanum*.
RI = Retention index, TMTT = (3*E*,7*E*)–4,8,12-Trimethyltrideca-1,3,7,11-tetraene, DMNT = (*E*)–4,8-Dimethylnona-1,3,7-triene.

| Substance Name | Distributor | Article No | CAS No | Quantifier [m/z] | Qualifiers [m/z (height in %)] | RI |
|---|---|---|---|---|---|---|
| (*E*)–2-Hexenal | Aldrich | 132659–25 g | 6728-26-3 | 41 | 55.00 (82.67) 69.00 (72.87) | 1215 |
| (*E*)-α-Bergamotene | isobionics | - | - | 119 | 93.00 (95.07) 91.00 (38.14) | 1594 |
| (*E*)-β-Ocimene | Sigma-Aldrich | W353901-SAMPLE | 13877-91-3 | 93 | 91.00 (47.39) 79.00 (45.35) | 1248 |
| (*Z*)–1,5-octadien-3-ol | Givaudan | - | - | 57 | 70.00 (45.40) 55.00 (34.19) | 1473 |
| (*Z*)–3-Hexen-1-ol | Phyto Technology Laboratories | H4000 | 928-96-1 | 67 | 82.00 (46.66) 55.00 (42.27) | 1378 |
| (*Z*)- 3-Hexen-1-ol, acetate | Sigma-Aldrich | 74597–1 ml | 3681-71-8 | 43 | 67.00 (98.54) 82.00 (52.15) | 1317 |
| 1-Hexanol | Fluka | 52830 | 111-27-3 | 56 | 55.00 (47.00) 69.00 (28.00) | 1346 |
| 1-Octen-3-ol | Sigma-Aldrich | 05284–25 g | 3391-86-4 | 57 | 43.00 (24.99) 72.00 (16.88) | 1446 |
| 1,2,3-Trimethyl-benzene | Aldrich | T73202-5mL | 526-73-8 | 105 | 120.00 (50.09) 77.00 (11.00) | 1333 |
| 3-Octanone | Sigma-Aldrich | 136913–25 g | 106-68-3 | 57 | 71.00 (70.00) 99.00 (67.00) | 1255 |
| 3-Pentanol | Aldrich | P8025 | 584-02-1 | 59 | 41.00 (19.00) 58.00 (9.00) | 1102 |
| α-Humulene | PhytoLab | 83351–100 mg | 6753-98-6 | 93 | 80.00 (32.44) 121.00 (26.27) | 1679 |
| β-Caryophyllene | Sigma-Aldrich | 22075–5 ml-F | 87-44-5 | 93 | 133.00 (91.17) 69.00 (87.31) | 1606 |
| DMNT | trc | TRC-D475810 | 19945-61-0 | 69 | 79.00 (14.49) 81.00 (13.76) | 1310 |
| Germacrene D | Cayman Chemical Company | 26539 | 62235-06-7 | 161 | 105.00 (78.00) 119.00 (52.00) | 1710 |
| Hexyl acetate | Sigma-Aldrich | 25539–1 mL | 142-92-7 | 43 | 56.00 (77.00) 61.00 (34.00) | 1270 |
| Linalool | Aldrich | L2602-5g | 78-70-6 | 71 | 93.00 (82.13) 121.00 (24.00) | 1544 |
| Indole | Sigma-Aldrich | 442619 | 120-72-9 | 117 | 90.00 (45.80) 89.00 (31.87) | 2483 |
| TMTT | trc | TRC-T797630 | 62235-06-7 | 69 | 81.00 (43.14) 79.00 (11.34) | 1814 |

## Oviposition choice bioassays

### Plants

All plants were planted in plastic pots in black cotton soil in insect-proof greenhouses in Mbita, Kenya. Maize plants (SC Duma 43, Seed Co Limited, Nairobi, Kenya) were grown from seeds with the addition of fertilizer (3 g DAP 18-46-0, Yara East Africa LTD, Nairobi, Kenya) and used at the age of 2–4 weeks with five to seven fully-grown leaves. *Desmodium* plants were obtained from a push-pull field on the icipe Mbita campus and kept in pots without fertilizer.

### Fall armyworm moths

To form the FAW colony, wild individuals were collected in the counties Siaya, Kisumu, Migori, and Vihiga in Western Kenya. Larvae were fed on artificial diet based on soy flour, wheat germ, raw linseed oil, mineral mix, sugar, aureomycin, vitamins, agar, methyl parabene, sorbic acid, and calcium propionate (Article No 870-265-3747, Southland Products Inc, Lake Village AR, USA). Moths were fed on water and, in rare cases after emergence, on 10% honey solution. The colony was occasionally restocked with wild FAW moths. Each day after hatching, moths of both sexes were transferred to a cage to allow mating until the start of the experiment.

### Experimental procedure

Cages (100 x 60 x 60 cm) lined with wooden floors and ceilings and netted walls were placed in greenhouses that allowed air circulation through mesh side walls (for pictures of the real setup, see *Figure 4—figure supplement 2*, for exact positioning in the greenhouses, see *Figure 4—figure supplements 3 and 4*). Two plants were placed within the cage at the greatest distance possible (approximately 60 cm) in two opposite corners. For the control treatment, two maize plants were

placed in the cage, while the direct *Desmodium* treatment consisted of one maize plant and one *Desmodium* plant in the cage. In two additional treatments with indirect contact, two maize plants were placed in the cage, while one *Desmodium* plant was placed outside the cage unreachable for the FAW moths, but in proximity to one maize plant. Three female and two male moths aged 4–5 days were released approximately 1 hr before dusk and allowed to oviposit under natural conditions of L12:D12 for three consecutive nights. Eggs were collected in five different groups depending on the position in which they were laid: either on one of the two plants, close to one of the plants within a maximum of 20 cm distance, or on the cage further than 20 cm from any plant (labeled as 'No Decision'). Each treatment was repeated 19–21 times in total over five cycles, with up to five replicates per treatment and cycle.

### Egg count

Each batch of FAW eggs was collected with sticky tape to separate the layers and spread all eggs out to one dimension. Each egg batch was taped to white paper and photographed in a UV imaging system (Syngene, Cambridge, England) against UV light (312 nm) coming from underneath the paper. A script for semi-automatic counts in ImageJ (Version 1.54 f, National Institutes of Health, USA) was developed and used for counting the exact number of eggs per picture (see *Figure 4—figure supplement 5*). The code is available in the data and code repository.

## Dual-choice and no-choice assays

### Plants

All plants were planted from seeds without the addition of fertilizer in plastic pots in standard red soil mixed with manure (ratio of 2:1) at the icipe campus in insect-proof greenhouses in Nairobi, Kenya. Maize plants (SC Duma 43, Seed Co Limited, Nairobi, Kenya) were used at the age of 3–7 weeks with five to eight fully grown leaves. It was observed that maize plants seemed to grow more slowly in Nairobi than in Mbita, which could be explained by the generally lower temperatures in Nairobi. The seeds of *D. intortum* and *D. incanum* were collected from a push-pull field located on the icipe campus in Mbita. These seeds were subsequently propagated through the replanting of cuttings, which were periodically trimmed to ensure healthy growth.

### Fall armyworm moths

To form the FAW colony, wild individuals were collected in Central Kenya such as in the counties Kiambu, Muranga, and Embu and restocked with wild individuals every three months. Larvae were reared on maize leaves, while moths were only fed with water. The colony was occasionally restocked with wild FAW moths. Each day after hatching, moths of both sexes were transferred to a cage to allow mating until the start of the experiment. At least 2 hr prior to the experiment, female moths were separated from males and placed in the darkened experimental room for adjustment to temperature and humidity.

### Dual-choice assay: experimental procedure

The five upper leaves of a maize plant were carefully wrapped in a preheated (200 °C for at least 1 hr) and cooled roasting bag (Sainsbury's, London, UK) with the addition of approximately the same biomass of *Desmodium* on one side of a dual-choice assay setup (100 x 30 x 30 cm) and compared to a single maize plant wrapped in the same manner on the other side. Maize leaves were carefully curved without damaging them to fit into the bag, while the bag was secured to the bottom edge with little tension using plastic straps supplied with the oven bags. Control treatments were conducted with a single maize plant wrapped in this way on each side. Air was pushed with the help of a Volatile Collection Kit (B. J. Pye, Kings Walden, UK) through activated charcoal filters and into the roasting bag from the lower rim of the bag at a rate of 740–820 mL/min with a PTFE tube with a diameter of approximately 2 mm. A PTFE tube with a diameter of approximately 10 mm and a length of approximately 120 cm conveyed the plant volatiles from the top of the roasting bag to one side of the dual-choice assay. Inside the dual-choice assay, the air was pumped out from the center at the 0 cm mark resulting in an air stream at the air transfer tubing of both sides of the dual-choice assay setup. Airflow was measured each day on both sides using a portable flow meter (Vögtlin Instruments

GmbH, Switzerland) and ranged between 480 and 570 mL/min across different days, with a maximum difference of 4 mL/min between the two sides. The higher incoming airflow created overpressure in the bags, which prevented unfiltered air from entering the dual-choice system. One female moth at an age of 4–5 days was released through a hole in the center at 0 cm and her flight behavior was observed for 5 min. The dual-choice assay was separated into five segments of 20 cm each, and a time stamp was set every time the moth changed segments (See *Figure 5—figure supplement 1*). All experiments were conducted between 7 pm and 1:30 am under red light. Each trial was recorded on video using a mobile phone camera (Fairphone 4, 48 MP), which captured the dual-choice assay from a front view. The videos were re-watched for data acquisition and in cases where moths were barely visible, complemented with notes from the live observation. To simplify the data analysis, the two left segments (50–30 cm left and 30–10 cm left) were combined, as were the two right segments (10–30 cm right, 30–50 cm right). An equal number of repetitions for each treatment was conducted consecutively each night, with the order of treatments randomized. Changing each set of plants took on average 15 min, while the room was ventilated for at least 30 min with ambient air between treatment changes.

## No-choice assay: experimental procedure

The five upper leaves of a maize plant were carefully wrapped in a preheated (100 °C for at least 1 hr) and cooled roasting bag (Sealapack, Manchester, UK) with the addition of approximately the same biomass of *Desmodium* on one side of a no-choice assay setup (115 x 33 x 33 cm). Maize leaves were carefully curved without damaging them to fit into the bag, while the bag was secured to the bottom edge with little tension using plastic straps supplied with the oven bags. Maize treatments were conducted with one maize plant only, and control treatments were performed with an empty roasting bag. Air was drawn through activated charcoal filters and directed into the roasting bag via a PTFE tube (approximately 2 mm in diameter) at a flow rate of 1.5 L/min, with the air entering the bag from its lower rim. This process was facilitated using a Volatile Collection Kit (B. J. Pye, Kings Walden, UK). A PTFE tube was used, with a diameter of approximately 2 mm and a length of 120 cm, to transport plant volatiles from the top of the roasting bag to the source on the side of the no-choice assay. On the opposite side, air was evacuated at the 0 cm mark, creating a uniform air stream that flowed across the entire chamber of the no-choice assay at 30 cm/s. The higher incoming airflow created overpressure in the bags, which prevented unfiltered air from entering the dual-choice system. One female moth aged 4–5 days was released through a hole in the center at 0 cm and its flight behavior was observed for 5 min. The initial and final landing distances were noted, while the last movement was determined by videos that were recorded for each repetition. In cases where moths were not visible or only partially visible, we recorded their last clearly observed position as the final location, assuming any significant movement would have been captured on video. All experiments were carried out between 9 pm and 1 am under red light conditions. The same set of plants was used for five repetitions, with a different moth individual introduced for each repetition. Treatments were carried out on separate evenings to avoid any cross-contamination or carryover effects. All repetitions of a single treatment were conducted on the same evening, except for the maize (alone) treatment, which was carried out across two evenings.

## Statistical analysis

All statistical analysis was performed in RStudio (R version 4.5.0, RStudio version 2024.12.1). Mixed models were used to determine significant effects of FAW egg positions in the oviposition bioassays, as well as the behavior of FAW moths in the dual-choice and no-choice assays. In the oviposition and dual-choice bioassays, control treatments with two maize plants were included. Therefore, the control replications were included twice in the dataset, once with the egg count data or the duration of stay of both sides interchanged. For the oviposition bioassay, the egg counts on the maize or the *Desmodium* plant were compared amongst all treatments, with the inclusion of the greenhouse, the cage (combination of the cage group and the cage number), the start date, and the replication number as random effects. For the dual-choice assay, the duration of stay closer to the maize or the *Desmodium* plants was compared amongst all treatments, with the inclusion of the date and the replication number as random effects. For the no-choice assay, the initial flight distance and the landing flight distance were compared amongst all treatment, with the inclusion of the date and the replication as random effects.

## Acknowledgements

This research was supported by the European Union's Horizon 2020 research and innovation programme under grant agreement No. 861998. We are very thankful for experimental assistance from Silas Ouko, Chrispin Onyango, Nashon Opiyo, Michael Machongo, and Amos Mwangangi.

## Additional information

### Competing interests
Bernhard Schmid: Reviewing editor, *eLife*. Meredith C Schuman: Senior editor, eLife. The other authors declare that no competing interests exist.

### Funding

| Funder | Grant reference number | Author |
|---|---|---|
| European Commission | 10.3030/861998 | Daria M Odermatt<br>Frank Chidawanyika<br>Daniel M Mutyambai<br>Collins O Onjura<br>Amanuel Tamiru<br>Meredith C Schuman |

The funders had no role in study design, data collection and interpretation, or the decision to submit the work for publication.

### Author contributions

Daria M Odermatt, Conceptualization, Data curation, Formal analysis, Validation, Investigation, Visualization, Methodology, Writing – original draft, Writing – review and editing; Frank Chidawanyika, Conceptualization, Resources, Supervision, Funding acquisition, Methodology, Project administration, Writing – review and editing; Daniel M Mutyambai, Supervision, Writing – review and editing; Bernhard Schmid, Formal analysis, Writing – review and editing; Luiz A Domeignoz Horta, Supervision, Visualization, Writing – review and editing; Collins O Onjura, Investigation, Writing – review and editing; Amanuel Tamiru, Resources, Supervision, Writing – review and editing; Meredith C Schuman, Conceptualization, Resources, Supervision, Funding acquisition, Methodology, Writing – original draft, Project administration, Writing – review and editing

### Author ORCIDs

Daria M Odermatt ⓘ https://orcid.org/0009-0002-9246-0209
Frank Chidawanyika ⓘ https://orcid.org/0000-0002-4601-768X
Daniel M Mutyambai ⓘ https://orcid.org/0000-0003-0556-1707
Bernhard Schmid ⓘ https://orcid.org/0000-0002-8430-3214
Luiz A Domeignoz Horta ⓘ https://orcid.org/0000-0003-4618-6253
Collins O Onjura ⓘ https://orcid.org/0009-0008-8165-0369
Amanuel Tamiru ⓘ https://orcid.org/0000-0003-2735-0855
Meredith C Schuman ⓘ https://orcid.org/0000-0003-3159-3534

Reviewer #2 (Public review): https://doi.org/10.7554/eLife.100981.3.sa1
Author response https://doi.org/10.7554/eLife.100981.3.sa2

## Additional files

### Supplementary files
MDAR checklist

Supplementary file 1. Overview of push–pull studies on FAW and detailed list of detected volatile compounds. Sheet "Overview Studies": Study = Abbreviated name of the study. The full citations can be found at the bottom of the table. Plants studied = List of plant species investigated in the corresponding study. Set-up = Type of experimental design, ranging from field observations

to bioassays and volatile sampling. Method = More detailed categorization of the experimental approach used. Metric = Measurement unit or parameter used in the method. Conclusions = Main outcome of the experiment. Sheet "Volatile Substances": Molecule = Name of the molecule found in the *Desmodium* species. Sum Formula = Sum formula of the corresponding substance. Reported for *Desmodium intortum* = Studies that reported the corresponding substance in relation to *D. intortum*. To our knowledge, no volatile profile of *D. incanum* has been reported so far. The full citations can be found at the bottom of the table. Classification = Classification of the substance based on literature. Classification Literature = Literature used for the classification of the substance. The full citations can be found at the bottom of the table. Qualitative occurrence in ambient control OR field *D. incanum* OR field *D. intortum* = Proportion of repetitions in which the substance was detected relative to the total number of repetitions. Mean quantitative occurrence in ambient control OR field *D. incanum* OR field *D. intortum* (zero values removed) = Mean $\log_{10}$-area calculated from samples in which the substance was detected; zero values from non-detects were excluded. Mean quantitative occurrence in ambient control OR field *D. incanum* OR field *D. intortum* (zero values included) = Mean $\log_{10}$-area calculated across all samples, including zero values. Reference CAS No = CAS number assigned to the commercially available substance. Reference Purity = Description of whether an isomer mixture or a pure isomer was used as a reference for comparison of the retention time and the mass spectra.

### Data availability

Raw and processed data of volatile sampling and the bioassays are stored on Zenodo. Statistical analysis were conducted in R / Rstudio and all code is available on GitHub (copy archived at *Odermatt, 2025*).

The following dataset was generated:

| Author(s) | Year | Dataset title | Dataset URL | Database and Identifier |
|---|---|---|---|---|
| Odermatt DM | 2024 | Raw data for publication "Desmodium volatiles in "push-pull" cropping systems and protection against the fall armyworm, *Spodoptera frugiperda*" | https://doi.org/10.5281/zenodo.11633889 | Zenodo, 10.5281/zenodo.11633889 |

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

## Appendix 1

### Pretests: moth activity window

Pre-tests were carried out to determine the activity window of the FAW moths, as it was important for deciding the time window of the headspace sampling of volatiles that are relevant for the FAW moths. The FAW moth is reported to be nocturnal (*Sparks, 1979*) and was checked regularly from dusk until around midnight. In the first hours only little activity was observed, while toward midnight, more flight and mating activity was observed. As a result, the moth activity could not be limited to the few hours of the evening, and volatile sampling was proceeded from 7 pm to 6:45 am covering all dark hours.

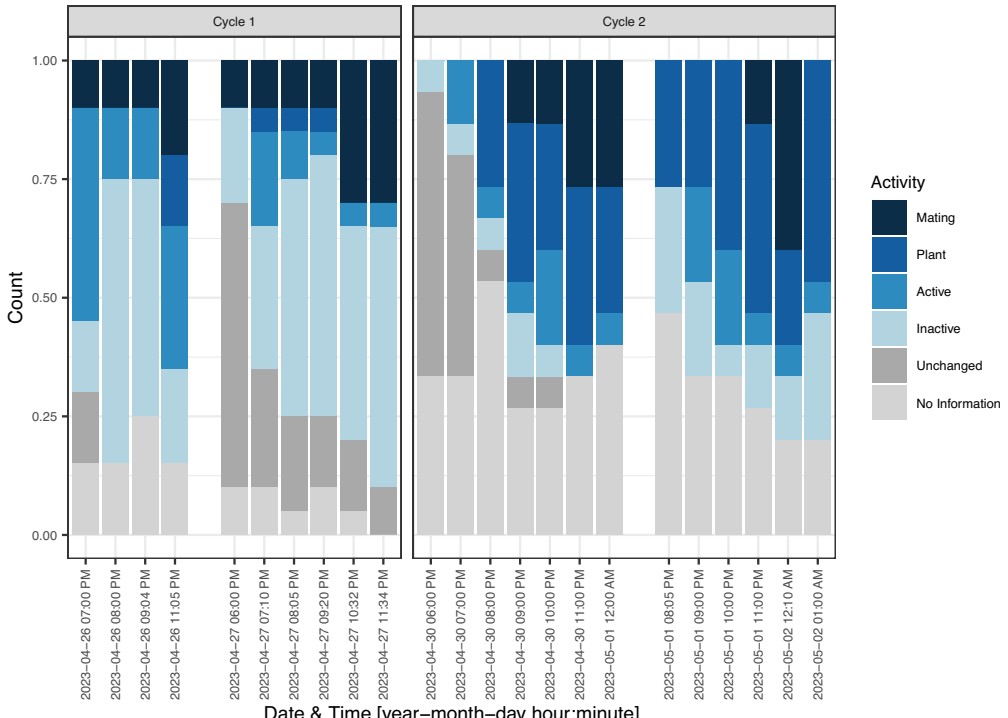

**Appendix 1—figure 1.** Stacked bar plot displaying the moth activity between 6 pm and 1 am. Several pairs of moths were observed mating for several hours up until 1 pm. Explanation color code: Mating = Two moths mate, Plant = Moth sits on the plant and therefore might oviposit, Active = Moth actively moves or flies, Inactive = Moth does not move and sits on the frame or the net, Unchanged = Moth does not move from the starting point, No Information = Moth could not be found and therefore no information was obtained about its activity.

