## [Editor Report · eLife Assessment]

Research on push-pull systems has often focused on controlled environments, leaving significant gaps in our understanding of how these systems function under real-world conditions. This **important** and **solid** study makes a substantial contribution by investigating the volatile emissions and behavioral effects of Desmodium in natural and semi-field contexts which offer insights of broad interest for sustainable agriculture and pest management. While the authors rightly acknowledge some remaining limitations, the revised manuscript now provides a well-supported and transparent assessment of the ecological role of Desmodium volatiles in push-pull systems.

---

## [Referee Report · Reviewer #2 (Public review)]

Based on the controversy of whether the Desmodium intercrop emits bioactive volatiles that repel the fall armyworm, the authors conducted this study to assess the effects of the volatiles from Desmodium plants in the push-pull system on behavior of FAW oviposition. This topic is interesting and the results are valuable for understanding the push-pull system for the management of FAW, the serious pest. The methodology used in this study is valid, leading to reliable results and conclusions. I just have a few concerns and suggestions for the improvement of this paper:

(1) The volatiles emitted from D. incanum were analyzed and their effects on the oviposition behavior of FAW moth were confirmed. However, it would be better and useful to identify the specific compounds that are crucial for the success of the push-pull system.

(2) That would be good to add "symbols" of significance in Figure 4 (D).

(3) Figure A is difficult for readers to understand.

(4) It will be good to deeply discuss the functions of important volatile compounds identified here with comparison with results in previous studies in the discussion better.

Comments on revisions:

The authors addressed all my concerns, and I believe that the current version is appropriate for publication.

---

## [Author Response]

The following is the authors’ response to the original reviews.

**Reviewer #1 (Public review):**
Summary:The manuscript of Odermatt et al. investigates the volatiles released by two species of *Desmodium* plants and the response of herbivores to maize plants alone or in combination with these species. The results show that *Desmodium* releases volatiles in both the laboratory and the field. Maize grown in the laboratory also released volatiles, in a similar range. While female moths preferred to oviposit on maize, the authors found no evidence that *Desmodium* volatiles played a role in lowering attraction to or oviposition on maize.Strengths:The manuscript is a response to recently published papers that presented conflicting results with respect to whether *Desmodium* releases volatiles constitutively or in response to biotic stress, the level at which such volatiles are released, and the behavioral effect it has on the fall armyworm. These questions are relevant as *Desmodium* is used in a textbook example of pest-suppressive sustainable intercropping technology called push-pull, which has supported tens of thousands of smallholder farmers in suppressing moth pests in maize. A large number of research papers over more than two decades have implied that *Desmodium* suppresses herbivores in push-pull intercropping through the release of large amounts of volatiles that repel herbivores. This premise has been questioned in recent papers. Odermatt et al. thus contribute to this discussion by testing the role of odors in oviposition choice. The paper confirms that ovipositing FAW preferred maize, and also confirmed that odors released from *Desmodium* appeared not important in their bioassays.The paper is a welcome addition to the literature and adds quality headspace analyses of *Desmodium* from the laboratory and the field. Furthermore, the authors, some of whom have since long contributed to developing push-pull, also find that *Desmodium* odors are not significant in their choice between maize plants. This advances our knowledge of the mechanisms through which push-pull suppresses herbivores, which is critically important to evolving the technique to fit different farming systems and translating this mechanism to fit with other crops and in other geographical areas.

Thank you for your careful assessment of our manuscript.

Weaknesses:Below I outline the major concerns:(1) Clear induction of the experimental plants, and lack of reflective discussion around this: from literature data and previous studies of maize and *Desmodium*, it is clear that the plants used in this study, particularly the *Desmodium*, were induced. Maize appeared to be primarily manually damaged, possibly due to sampling (release of GLV, but little to no terpenoids, which is indicative of mostly physical stress and damage, for example, one of the coauthor's own paper Tamiru et al. 2011), whereas *Desmodium* releases a blend of many compounds (many terpenoids indicative of herbivore induction). Erdei et al. also clearly show that under controlled conditions maize, silver leaf and green leaf Desmodium release volatiles in very low amounts. While the condition of the plants in Odermatt et al. may be reflective of situations in push-pull fields, the authors should elaborate on the above in the discussion (see comments) such that the readers understand that the plant's condition during the experiments. This is particularly important because it has been assumed that *Desmodium* releases typical herbivore-induced volatiles constitutively, which is not the case (see Erdei et al. 2024). This reflection is currently lacking in the manuscript.

We acknowledge the need for a more reflective discussion on the possible causes of volatile emission due to physical damage. Although the field plants were carefully handled, it is possible that some physical stress may have contributed to the release of volatiles, such as green leaf volatiles (GLVs). We ensured the revised manuscript reflects this nuanced interpretation (lines 282 – 286). However, we also explained more clearly that our aim was to capture the volatile emission of plants used by farmers under realistic conditions and moth responses to these plants, not to be able to attribute the volatile emission to a specific cause (lines 115 – 117). We revised relevant passages throughout the results and discussion to ensure that we do not make any claims about the reason for volatile emissions, and that our claims regarding these plants and their headspace being representative of the system as practiced by farmers are supported. In the revised manuscript we provide a new supplementary table S2 that additionally shows the classification of the identified substances, which also shows that the majority of the substances that were found in the headspace of the sampled plants of *Desmodium intortum* or *Desmodium incanum* are monoterpenes, sesquiterpenes, or aromatic compounds, and not GLVs (that are typically emitted following damage).

(2) Lack of controls that would have provided context to the data: The experiments lack important controls that would have helped in the interpretation:2a The authors did not control the conditions of the plants. To understand the release of volatiles and their importance in the field, the authors should have included controlled herbivory in both maize and *Desmodium*. This would have placed the current volatile profiles in a herbivory context. Now the volatile measurements hang in midair, leading to discussions that are not well anchored (and should be rephrased thoroughly, see eg lines 183-188). It is well known that maize releases only very low levels of volatiles without abiotic and biotic stressors. However, this changes upon stress (GLVs by direct, physical damage and eg terpenoids upon herbivory, see above). Erdei et al. confirm this pattern in *Desmodium*. Not having these controls, means that the authors need to put the data in the context of what has been published (see above).

We appreciate this concern. Our study aimed to capture the real-world conditions of push-pull fields, where *Desmodium* and maize grow in natural environments without the direct induction of herbivory for experimental purposes (lines 115 – 117). We agree that in further studies it would be important to carry out experiments under different environmental conditions, including herbivore damage. However, this was not within the scope of the present study.

2b It would also have been better if the authors had sampled maize from the field while sampling *Desmodium*. Together with the above point (inclusion of herbivore-induced maize and *Desmodium*), the levels of volatile release by *Desmodium* would have been placed into context.

We acknowledge that sampling maize and other intercrop plants, such as edible legumes, alongside *Desmodium* in the push-pull field would have allowed us to make direct comparisons of the volatile profiles of different plants in the push-pull system under shared field conditions. Again, this should be done in future experiments but was beyond the scope of the present study. Due to the amount of samples we could handle given cost and workload, we chose to focus on *Desmodium* because there is much less literature on the volatile profiles of field-grown *Desmodium* than maize plants in the field: we are aware of one study attempting to measure field volatile profiles from *Desmodium intortum* (Erdei et al. 2024) and no study attempting this for *Desmodium incanum*. We pointed out this justification for our focus on *Desmodium* in the manuscript (lines 435 - 439). Additionally, we suggested in the discussion that future studies should measure volatile profiles from all plants commonly used in push-pull systems alongside *Desmodium* (lines 267 – 269).

2c To put the volatiles release in the context of push-pull, it would have been important to sample other plants which are frequently used as intercrop by smallholder farmers, but which are not considered effective as push crops, particularly edible legumes. Sampling the headspace of these plants, both 'clean' and herbivore-induced, would have provided a context to the volatiles that *Desmodium* (induced) releases in the field - one would expect unsuccessful push crops to not release any of these 'bioactive' volatiles (although 'bioactive' should be avoided) if these odors are responsible for the pest suppressive effect of *Desmodium*. Many edible intercrops have been tested to increase the adoption of push-pull technology but with little success.

We very much agree that such measurements are important for the longer-term research program in this field. But again, for the current study this would have exploded the size of the required experiment. Regarding bioactivity, we have been careful to use the phrase "potentially bioactive" solely when referring to findings from the literature (lines 99–103), in order to avoid making any definitive claims about our own results.

Because of the lack of the above, the conclusions the authors can draw from their data are weakened. The data are still valuable in the current discussion around push-pull, provided that a proper context is given in the discussion along the points above.

We think our revisions made the specific aims of this study more explicit and help to avoid misleading claims.

(3) 'Tendency' of the authors to accept the odor hypothesis (i.e. that *Desmodium* odors are responsible for repelling FAW and thereby reduce infestation in maize under push-pull management) in spite of their own data: The authors tested the effects of odor in oviposition choice, both in a cage assay and in a 'wind tunnel'. From the cage experiments, it is clear that FAW preferred maize over *Desmodium*, confirming other reports (including Erdei et al. 2024). However, when choosing between two maize plants, one of which was placed next to *Desmodium* to which FAW has no tactile (taste, structure, etc), FAW chose equally. Similarly in their wind tunnel setup (this term should not be used to describe the assay, see below), no preference was found either between maize odor in the presence or absence of *Desmodium*. This too confirms results obtained by Erdei et al. (but add an important element to it by using *Desmodium* plants that had been induced and released volatiles, contrary to Erdei et al. 2024). Even though no support was found for repellency by *Desmodium* odors, the authors in many instances in the manuscript (lines 30-33, 164-169, 202, 279, 284, 304-307, 311-312, 320) appear to elevate non-significant tendencies as being important. This is misleading readers into thinking that these interactions were significant and in fact confirming this in the discussion. The authors should stay true to their own data obtained when testing the hypothesis of whether odors play a role in the pest-suppressive effect of push-pull.

We appreciate this feedback and agree that we may have overstated claims that could not be supported by strict significance tests. However, we believe that non-significant tendencies can still provide valuable insights. In the revised version of the manuscript, we ensured a clear distinction between statistically significant findings and non-significant trends and remove any language that may imply stronger support for the odor hypothesis than what the data show in all the lines that were mentioned.

(4) Oviposition bioassay: with so many assays in close proximity, it is hard to certify that the experiments are independent. Please discuss this in the appropriate place in the discussion.

We have pointed this out in the submitted manuscript in lines 275 – 279. Furthermore, we included detailed captions to figure 4 - supporting figure 3 & figure 4 - supporting figure 4. We are aware that in all such experiments there is a danger of between-treatment interference, which we pointed out for our specific case. We stated that with our experimental setup we tried to minimize interference between treatments by spacing and temporal staggering. We would like to point out that this common caveat does not invalidate experimental designs when practicing replication and randomization. We assume that insects are able to select suitable oviposition sites in the background of such confounding factors under realistic conditions.

(5) The wind tunnel has a number of issues (besides being poorly detailed):5a. The setup which the authors refer to as a 'wind tunnel' does not qualify as a wind tunnel. First, there is no directional flow: there are two flows entering the setup at opposite sides. Second, the flow is way too low for moths to orient in (in a wind tunnel wind should be presented as a directional cue. Only around 1.5 l/min enters the wind tunnel in a volume of 90 l approximately, which does not create any directional flow. Solution: change 'wind tunnel' throughout the text to a dual choice setup /assay.)

We agree with these criticisms and changed the terminology accordingly from ‘wind tunnel’ to ‘dual choice assay’. We have now conducted an additional experiment which we called ‘no-choice assay’ that provides conditions closer to a true wind tunnel. The setup of the added experiment features an odor entry point at only one side of the chamber to create a more directional airflow. Each treatment (maize alone, maize + *D. intortum*, maize + *D. incanum*, and a control with no plants) was tested separately, with only one treatment conducted per evening to avoid cross-contamination, as described in the methods section of the no-choice assay.

5b. There is no control over the flows in the flight section of the setup. It is very well possible that moths at the release point may only sense one of the 'options'. Please discuss this.

We added this to the discussion (lines 369 – 374). The new no-choice assays also address this concern by using a setup with laminar flow.

5c. Too low a flow (1,5 l per minute) implies a largely stagnant air, which means cross-contamination between experiments. An experiment takes 5 minutes, but it takes minimally 1.5 hours at these flows to replace the flight chamber air (but in reality much longer as the fresh air does not replace the old air, but mixes with it). The setup does not seem to be equipped with e.g. fans to quickly vent the air out of the setup. See comments in the text. Please discuss the limitations of the experimental setup at the appropriate place in the discussion.

We added these limitations to the discussion and addressed these concerns with new experiments (see answer 5a).

5d. The stimulus air enters through a tube (what type of tube, diameter, length, etc) containing pressurized air how was the air obtained into bags (type of bag, how is it sealed?), and the efflux directly into the flight chamber (how, nozzle?). However, it seems that there is no control of the efflux. How was leakage prevented, particularly how the bags were airtight sealed around the plants?

We added the missing information to the methods and provided details about types of bags, manufacturers, and pre-treatments in the method section. In short, PTFE tubes connected bagged plants to the bioassay setup and air was pumped in at an overpressure, so leakage was not eliminated but contamination from ambient air was avoided.

5e. The plants were bagged in very narrowly fitting bags. The maize plants look bent and damaged, which probably explains the GLVs found in the samples. The *Desmodium* in the picture (Figure 5 supplement, which we should assume is at least a representative picture?) appears to be rather crammed into the bag with maize and looks in rather poor condition to start with (perhaps also indicating why they release these volatiles?). It would be good to describe the sampling of the plants in detail and explain that the way they were handled may have caused the release of GLVs.

We included a more detailed description of the plant handling and bagging processes to the methods to clarify how the plants were treated during the dual-choice and the no-choice assays reported in the revised manuscript. We politely disagree that the maize plants were damaged and the *Desmodium* plants not representative of those encountered in the field. The plants were grown in insect-proof screen houses to prevent damage by insects and carefully curved without damaging them to fit into the bag. The *Desmodium* plant pictured was *D. incanum*, which has sparser foliage and smaller leaves than *D. intortum*.

(6) Figure 1 seems redundant as a main figure in the text. Much of the information is not pertinent to the paper. It can be used in a review on the topic. Or perhaps if the authors strongly wish to keep it, it could be placed in the supplemental material.

We think that Figure 1 provides essential information about the push-pull system and the FAW. To our knowledge, this partly contradictory evidence so far has not been synthesized in the literature. We realize that such a figure would more commonly be provided in a review article, but we do not think that the small number of studies on this topic so far justify a stand-alone review. Instead, the introduction to our manuscript includes a brief review of these few studies, complemented by the visual summary provided in Figure 1 and a detailed supplementary table.

**Reviewer #2 (Public review):**
Based on the controversy of whether the *Desmodium* intercrop emits bioactive volatiles that repel the fall armyworm, the authors conducted this study to assess the effects of the volatiles from *Desmodium* plants in the push-pull system on behavior of FAW oviposition. This topic is interesting and the results are valuable for understanding the push-pull system for the management of FAW, the serious pest. The methodology used in this study is valid, leading to reliable results and conclusions. I just have a few concerns and suggestions for improvement of this paper:(1) The volatiles emitted from D. incanum were analyzed and their effects on the oviposition behavior of FAW moth were confirmed. However, it would be better and useful to identify the specific compounds that are crucial for the success of the push-pull system.

We fully agree that identifying specific volatile compounds responsible for the push-pull effect would provide valuable insights into the underlying mechanisms of the system. However, the primary focus of this study was to address the still unresolved question whether *Desmodium* emits detectable or “significant” amounts of volatiles at all under field conditions, and the secondary aim was to test whether we could demonstrate a behavioral effect of *Desmodium* headspace on FAW moths. Before conducting our experiments, we carefully considered the option of using single volatile compounds and synthetic blends in bioassays. We decided against this because we judged that the contradictory evidence in the literature was not a sufficient basis for composing representative blends. Furthermore, we think it is an important first step to test f. or behavioral responses to the headspaces of real plants. We consider bioassays with pure compounds to be important for confirmation and more detailed investigation in future studies. There was also contradictory evidence in the literature regarding moth responses to plants. We thus opted to focus on experiments with whole plants to maintain ecological relevance.

(2) That would be good to add "symbols" of significance in Figure 4 (D).

We report the statistical significance of the parameters in Figure 4 (D) in Table 3, which shows the mixed model applied for oviposition bioassays. While testing significance between groups is a standard approach, we used a more robust model-based analysis to assess the effects of multiple factors simultaneously. We provided a cross-reference to Table 3 from the figure description of Figure 4 (D) for readers to easily find the statistical details.

(3) Figure A is difficult for readers to understand.

Unfortunately, it is not entirely clear which specific figure is being referred to as "Figure A" in this comment. We tried to keep our figures as clear as possible.

(4) It will be good to deeply discuss the functions of important volatile compounds identified here with comparison with results in previous studies in the discussion better.

Our study does not provide strong evidence that specific volatiles from *Desmodium* plants are important determinants of FAW oviposition or choice in the push-pull system. Therefore, we prefer to refrain from detailed discussions of the potential importance of individual compounds. However, in the revised version, we provide an additional table S2 which identifies the overlap with volatiles previously reported from *Desmodium*, as only the total numbers are summarized in the discussion of the submitted paper.

**Recommendations for the authors:**

**Reviewer #1 (Recommendations for the authors):**
The points raised are largely self-explanatory as to what needs to be done to fully resolve them. At a minimum the text needs to be seriously revised to:(1) reflect the data obtained.(2) reflect on the limitations of their experimental setup and data obtained.(3) put the data obtained and its limitations in what these tell us and particularly what not. Ideally, additional headspace measurements are taken, including from herbivory and 'clean' maize and *Desmodium* (in which there is better control of biotic and abiotic stress), as well as other crops commonly planted as companion crops with maize (but none of them reducing pest pressure).

Thank you for this summary. Please see our detailed responses above.

In addition to the main points of critique provided above, I have provided additional comments in the text (https://elife-rp.msubmit.net/elife-rp_files/2024/07/18/00134767/00/134767_0_attach_28_25795_convrt.pdf). These elaborate on the above points and include some new ones too. These are the major points of critique, which I hope the authors can address.

Thank you very much for these detailed comments.

**Reviewer #2 (Recommendations for the authors):**
It is important to note that the original push-pull system was developed against stemborers and involved Napier grass (still used) around the field, which attracts stemborer moths, and Molasses grass as the intercrop that repels the moths and attracts parasitoids. Later, Molasses grass was replaced by desmodiums because it is a legume that fixes nitrogen and therefore can increase nitrate levels in the soil, but most importantly because it prevents germination of the parasitic Striga weed. The possible repellent effect of desmodium on pests and attraction of natural enemies was never properly tested but assumed, probably to still be able to use the push-pull terminology. This "mistake" should be recognized here and in future publications. It is a real pity that the controversy over the repellent effect of desmodium distracts from the amazing success of the push-pull system, also against the fall armyworm.

We thank the reviewer for pointing out these issues, which are part of the reason for our Figure 1 and why we would like to keep it. We have described this development of the system in the introduction to better present the push-pull system. Our aim in Figure 1 and Table S1 is to highlight both the evidence of the system's success, and the gaps in our understanding, regarding specifically control of damage from the FAW.